# In situ evidence of the magnetospheric cusp of Jupiter from Juno spacecraft measurements

Y. Xu [1,2,3], C. S. Arridge[3], Z. H. Yao [1,2,4,5] ✉, B. Zhang[4], L. C. Ray [3], S. V. Badman [3], W. R. Dunn[5], R. W. Ebert [6,7], J. J. Chen [4], F. Allegrini[6,7], W. S. Kurth [8], T. S. Qin[4], J. E. P. Connerney [9,10], D. J. McComas [11], S. J. Bolton[6] & Y. Wei [1,2]

The magnetospheric cusp connects the planetary magnetic field to interplanetary space, offering opportunities for charged particles to precipitate to or escape from the planet. Terrestrial cusps are typically found near noon local time, but the characteristics of the Jovian cusp are unknown. Here we show direct evidence of Jovian cusps using datasets from multiple instruments onboard Juno spacecraft. We find that the cusps of Jupiter are in the dusk sector, which is contradicting Earth-based predictions of a near-noon location. Nevertheless, the characteristics of charged particles in the Jovian cusps resemble terrestrial and Saturnian cusps, implying similar cusp microphysics exist across different planets. These results demonstrate that while the basic physical processes may operate similarly to those at Earth, Jupiter's rapid rotation and its location in the heliosphere can dramatically change the configuration of the cusp. This work provides useful insights into the fundamental consequences of star-planet interactions, highlighting how planetary environments and rotational dynamics influence magnetospheric structures.

Global dipole magnetic fields commonly exist at solar system planets, with two exceptions (i.e., Venus and Mars). These planetary magnetic fields are persistently compressed by the high-speed solar wind, forming planetary magnetospheres. These magnetospheres vary significantly between planets. For instance, the energy released during substorms at Mercury is much greater than that measured at Earth[1]; at Jupiter, the rapid rotation rate of the planet, in addition to plasma sources embedded within the magnetosphere, results in a complex interaction with the solar wind[2–7], in contrast with the less complicated Dungey-driven magnetosphere prevalent at Earth.

Magnetic reconnection often occurs when at least one component of the planetary magnetic field and the interplanetary magnetic field align in an antiparallel configuration. During this reconnection process, magnetic field lines near the poles are directly connected to the interplanetary magnetic field, allowing solar wind and magnetosheath particles to enter the magnetosphere. The region where newly reconnected field lines allow plasma to enter the magnetosphere is known as the magnetospheric cusp[8–12]. The particles present within the cusp region contribute significantly to various dynamic processes occurring in the magnetosphere, ionosphere, and upper

[1]Key Laboratory of Earth and Planetary Physics, Institute of Geology and Geophysics, Chinese Academy of Sciences, Beijing, China. [2]College of Earth and Planetary Sciences, University of Chinese Academy of Sciences, Beijing, China. [3]Department of Physics, Lancaster University, Lancaster, UK. [4]NWU-HKU Joint Centre of Earth and Planetary Sciences, Department of Earth Sciences, University of Hong Kong, Hong Kong SAR, China. [5]Department of Physics and Astronomy, University College London, London, UK. [6]Southwest Research Institute, San Antonio, TX, USA. [7]Department of Physics and Astronomy, University of Texas at San Antonio, San Antonio, TX, USA. [8]Department of Physics and Astronomy, University of Iowa, Iowa City, IA, USA. [9]Space Research Corporation, Annapolis, MD, USA. [10]NASA/Goddard Space Flight Center, Greenbelt, MD, USA. [11]Department of Astrophysical Sciences, Princeton University, Princeton, NJ, USA. ✉e-mail: yaozh@hku.hk

atmosphere, including storms[13], substorms[14,15], and auroras[16,17]. The role that the solar wind plays in driving the Jovian magnetosphere has been a topic of active debate[3–7,18]. It is, therefore, crucial to improve our understanding of the characteristics of the cusp region and comprehend their implications for solar wind-magnetosphere coupling. Since the main auroras at Earth and Saturn are located close to the open-closed field line boundary, any auroras associated with the cusp are seen close to these main emissions[16]. Jupiter's main auroral emissions, on the other hand, are thought to be generated far from the open-closed boundary; thus, any cusp-related auroras should be located within the polar cap[5]. However, in contrast to Earth's typically dark polar regions, Jupiter exhibits bright and persistent polar auroras, which suggests that there are reconnection sites at unusual places connecting to the Jovian polar regions, hinting at differing cusp structures between the two planets[4,19–22].

The exact distribution and features of the cusp regions within Jupiter's magnetosphere remain unclear due to limited coverage by previous missions, which has hindered detailed investigation of Jupiter's cusps. Although no direct reports of the Jovian cusp have been made, several studies[23–25] have suggested cusp-related phenomena by utilizing observations from the Ulysses[26] spacecraft's 1992 flyby of Jupiter. During the encounter with Jupiter, evidence was found for open field lines linking to the polar region. Auroral hiss-like plasma waves were also detected during this period and deemed as potential evidence for the existence of a Jovian cusp[8]. However, due to the constraints of spacecraft instrumentation and data, there is a lack of conclusive evidence for the existence of the cusp regions[23]. The Juno spacecraft[27] offered a valuable opportunity to observe the cusp region at high latitudes on the dusk side at Jupiter.

In this study, we utilize Juno's plasma, magnetometer, and plasma wave datasets to present the comprehensive observation of the cusp region near the dusk side of Jupiter's magnetosphere.

## Results
### The identification processes for Jovian cusps
Earth's cusp has been comprehensively studied over the past few decades, and robust identification criteria have been developed[12,28–30]. Some cases of Saturn's cusp have also been identified and reported in detail[9–11]. To avoid potential confusion, this study follows the Earth and Saturn cusp definitions, i.e., a part of the magnetosphere in the vicinity of the polar region at high magnetic latitudes/invariant latitude, where a significant quantity of magnetosheath plasma is detected inside the magnetopause position[28–30]. In practice, magnetosheath-like electron distributions well inside the magnetopause in high latitudes often serve as the key identification criteria[10,31–33]. Furthermore, the cusp is the region where magnetosheath plasma and momentum enter the magnetosphere and are closely associated with magnetopause reconnection. Therefore, the ion dispersion feature due to reconnection-associated velocity filtering effects is also a useful feature for identifying the cusp, as extensively applied to identify the Earth[34–37] and Saturn's cusp[9–11]. Additionally, whistler-mode auroral hiss waves, combined with plasma data, are frequently utilized to identify the cusps of Earth[38–40] and Jupiter[8]. These waves serve as an auxiliary criterion to aid in the identification process. A detailed discussion of the different planetary cusp identification criteria is available in Supplementary Notes 1, 2 and Supplementary Tables 1, 2. Combining the above identification features, we obtained six typical Jupiter cusp events. In this paper, we focus on Case 1 and Case 2 (see Supplementary Note 3 for Jupiter's cases 3–6, and Supplementary Note 4 for Earth's and Saturn's cases). And we examine in situ measurements mainly from Juno's three instruments: the Juno Magnetic Field Investigation[41], the Jovian Auroral Distributions Experiment[42], and the Waves instrument[43]. The cusp event observations are presented in Figs. 1 and 2. Panels (a–e) in Figs. 1 and 2 show the magnetic field and plasma energy spectrum information. The electron pitch angle

distribution and wave emission are shown in panel (f) and panel (g) in Figs. 1 and 2, respectively. Moreover, Juno's footpoints, mapped to the polar region utilizing the JRM33 model[44] and the Connerney 2020 current sheet model[45] during the events, are displayed in Figs. 1h and 2h. Figures 1i and 2i show the trajectories of Juno around whole observations.

### Pre-dusk cusp structure
For the first cusp case on 27 June and 28 June, 2023, Juno was primarily positioned on the dusk side (about 17.8 LT) at high invariable latitudes (i.e., 79°–86°) shown in Fig. 1h, i. Before 2023 27 June, 19:13 UT, the spacecraft was mainly in the magnetosphere, as clearly demonstrated by the typical electron spectrum around 1000 eV[46], with the exceptions of two short periods (13:52-14:26 UT and 14:35-15:01 UT) when magnetosheath-like electrons were detected[47] along with protons with energies lower than magnetospheric population. Additionally, in these two regions, a distinct increase in auroral hiss[8,39] was observed, as marked by dashed lines. After 27 June, 19:13 UT, Juno rapidly entered into a region featuring with magnetosheath-like population, via a transition known as a boundary layer (BL) between magnetospheric and magnetosheath-like populations (Fig. 1c–g). The spacecraft stayed in the magnetosheath-like region for about 4 hours (Fig. 1c), connecting to the polar high-latitude region (Fig. 1h). The Juno spacecraft was about 54 $R_J$ away from Jupiter, which is well inside the magnetosphere[48]. Moreover, the magnetic field with little perturbation also confirms that Juno was in the magnetosphere rather than magnetosheath[47,49,50]. From 27 June, 20:22 UT to 28 June, 02:12 UT, a notable step-like proton dispersion[9,34] was observed (see Methods, subsection Location of Magnetic Reconnection, for details on the calculation of reconnection locations based on dispersion features) accompanied by intensified auroral hiss[8,39]. Based on the location (i.e., inside magnetopause and high latitude), plasma population (i.e., magnetosheath-like electrons and low-energy protons), proton dispersion, and auroral hiss waves, we thus confirm the observed structure is Jupiter's cusp.

A quantitative comparison of electron energy distributions for the cusp, magnetosphere, and magnetosheath is depicted in Fig. 3a, showing that the cusp distribution is similar to the one in the magnetosheath while with lower energy flux, akin to observations made for Saturn[9]. The stepped proton dispersion suggests possible pulsed magnetopause reconnection, akin to the observations reported in Saturn's cusp[9,11]. After 27 June 23:40 UT, within the cusp, there was an increase in the electron energy, accompanied by elevated fluxes of protons and heavy ions. The heavy ions exhibited enhancements at approximately 1000 eV and 10,000 eV, indicating a potential leakage from the magnetosphere. This diversity in plasma properties across different parts of the cusp region might suggest that the spacecraft traversed different flux tubes within the cusp, each carrying different particle streams[51]. Figure 1g highlights enhanced auroral hiss features within the three identified cusp regions, showing a strong correlation. And the auroral hiss is not confined solely to the cusp but also appears in the BL and some magnetospheric regions. This broader occurrence might result from the resonance cone angle's propagation of whistler-mode auroral hiss[52–54], which is tilted relative to the background magnetic field, allowing detection in adjacent or BL regions. The normalized electron pitch angle distribution in Fig. 1f displays the butterfly electron pitch angle distribution in the magnetosphere and its transition (BL) toward the cusp. But due to the lack of data coverage near 0° and 180°, the identification of field-aligned electrons associated with auroral hiss is challenging. As depicted in Fig. 1h, the field line tracing conducted in the System III coordinate system (which is fixed in planetary longitude and rotates with Jupiter) clearly shows the spacecraft's footpoints positioned well above the main auroral emissions. This observation further substantiates the high-latitude positioning of the cusp.

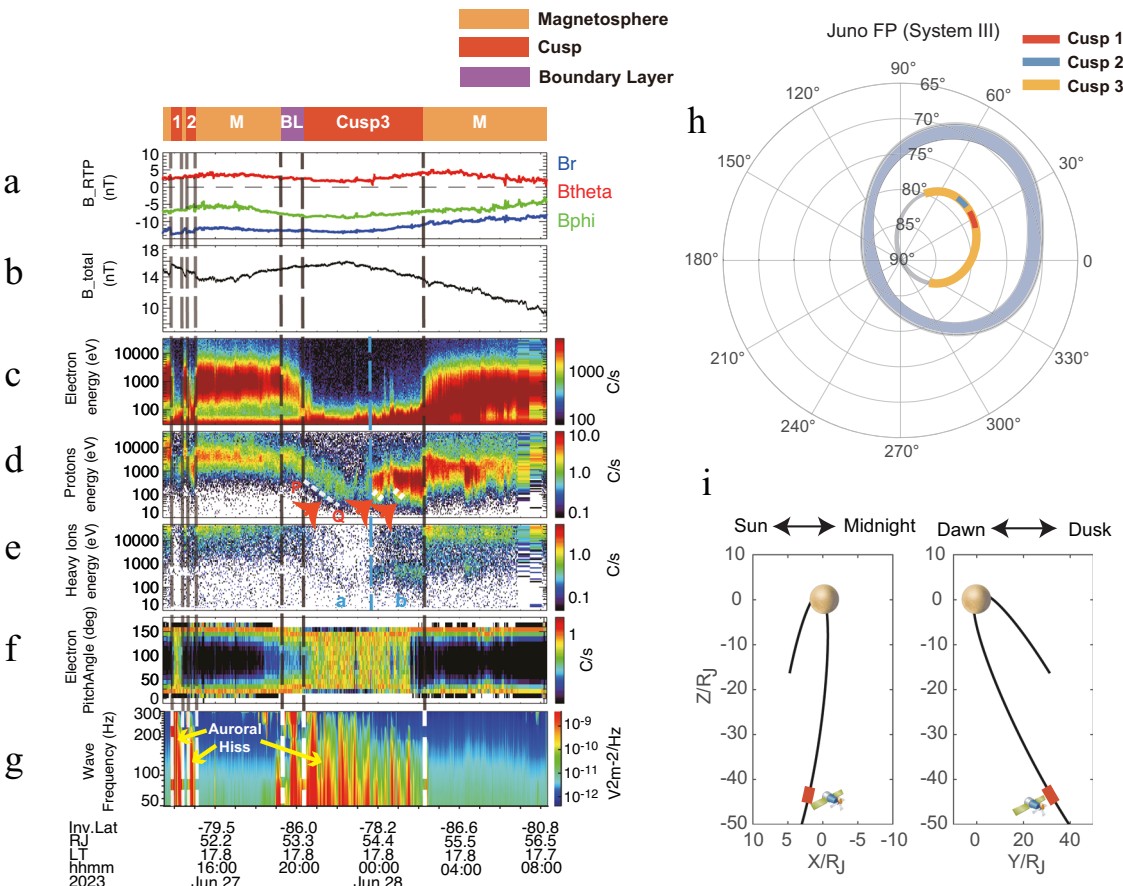

**Fig. 1 | The Jovian cusp observation (Case 1) on June 27 and June 28, 2023.**
**a** R-Theta-Phi magnetic field components in JSS (Jupiter-De-Spun-Sun) coordinate;
**b** The total magnetic field strength; **c** The electron energy spectrogram; Ion Energy
spectrogram for protons (**d**) and heavy ions (**e**), where heavy ions represent ions
with m/q in the range of 5 and 64[28]; **f** Pitch angle distribution for electrons which is
normalized at each time unit within energy ranges of 0.3 to 32 keV; **g** Plasma wave
observations in the frequency range 50 to 300 Hz. The different regions that the
spacecraft passes through are marked with different colors at the top and sepa-
rated by dashed lines. "M" is the magnetosphere, "C" is the cusp, "BL" is the
boundary layer. The red arrows and white dashed lines in panel (**d**) show the

dispersion. The yellow arrows in panel (**g**) indicate the enhanced auroral hiss fea-
tures. The blue dashed line demarcates the cusp into two regions, labeled as "**a**" and
"**b**", each characterized by different plasma properties. **h** Traced distribution of
spacecraft footprints before and after cusp observation in left-handed system III
coordinates. The blue regions are the main ovals, and the gray lines are the Juno
footprint trajectories from 26 June to 28 June, 2023. Colored lines are the Juno
footprint trajectories in cusps. **i** The position of the spacecraft around cusp
observations in JSS coordinate, red lines representing the time interval of the
cusp case.

## Post-dusk cusp structure

The spacecraft was positioned on the post-dusk side throughout the
observations, as indicated in Fig. 2i, with ~20.1 LT and a radial distance
of ~58 $R_J$. Given that Jupiter's magnetopause at 20 LT was predicted to
be roughly 130 $R_J$ to 200 $R_J$[55], it can be confirmed that Juno's location
was well within the magnetosphere, which is also confirmed by the
relatively stable magnetic field. Additionally, the spacecraft was at a
very high invariable latitude, as noted in Fig. 2h. The electron energy
spectrum in Fig. 2c shows that before 14 April, 2022, at 20:20 UT, the
spacecraft was primarily within the BL. Here, the electron energy
spectrum resembled that of the magnetosphere but with significantly
lower fluxes. Besides the BL features, Juno also detected two short
instances with clear magnetosheath-like distributions, and one short
period with typical magnetospheric features, as marked by the dashed
lines in Fig. 2 and the colored blocks at the top. From 20:20 UT to 21:30
UT on the same day when the magnetosheath-like electrons (~100 eV
enhancement) were detected, Juno also observed clear reversed pro-
ton dispersion features[11,36,56], as illustrated in Fig. 2d. Such ion reversed
dispersion features are common in the Earth's cusp with increasing
latitudes under northward interplanetary magnetic field (IMF)
conditions[17,36], with opposite directions for magnetic lines convection
motion and the spacecraft. It is important to note that no significant

magnetic field depressions were observed in the cusp region. Instead,
reductions only occur in the adjacent magnetosphere, as shown in
Fig. 2b. In some observations of Saturn cusp events, similar situations
were displayed where the magnetic depression is either absent or
insignificant[10,11,57]. Field-aligned electrons (>150° and <30° to magnetic
field lines) were detected both in the cusp and the regions immediately
ahead and behind it. These were accompanied by strong whistler-
mode auroral hiss waves, likely generated by the field-aligned electron
beams[53,58].

## The magnetic configuration associated with cusp

Figure 3c depicts a schematic illustrating the distribution of footprints
in magnetic coordinates for the open and closed magnetic field
regions, as well as the footprint distribution of the spacecraft, based on
the simulation results provided by Zhang et al.[2] (see Methods, sub-
section Simulation Information). According to Zhang et al.'s work[2], the
yellow and orange areas (corresponding to the polar aurora) represent
closed magnetic field regions. The blue region corresponds to the
open magnetic lines extending towards the far magnetotail, as indi-
cated by the blue lines in Fig. 3b. The green region represents the
footprint of the open magnetic field region associated with coupling to
the solar wind, as depicted by the green lines in Fig. 3b. This open

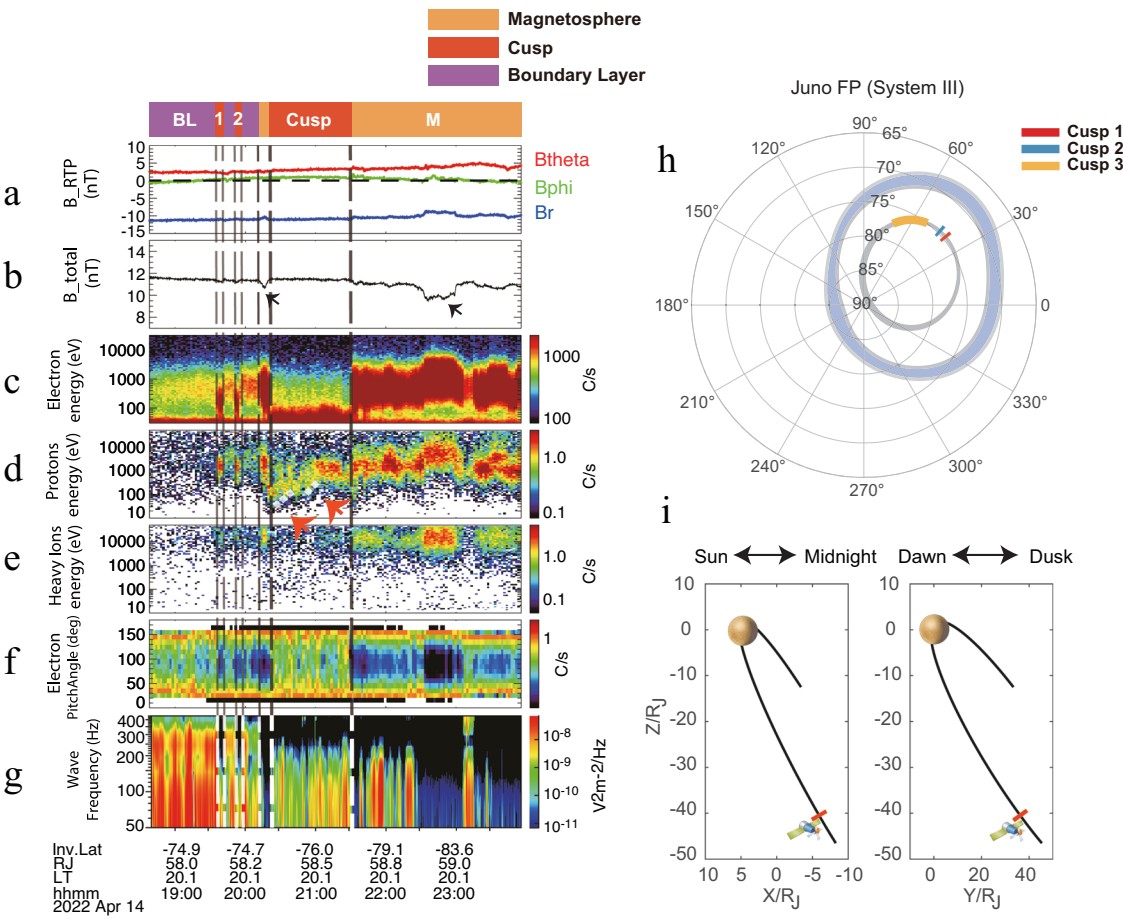

**Fig. 2 | The Jovian cusp Case 2 on 14 April, 2022.** This figure uses the same format as Fig. 1. The black arrow in panel (**b**) indicates a decrease in the magnetic field occurring within the magnetosphere, not in the cusp. The red arrows point to the reversed ion dispersion.

magnetic field region takes the form of a thin strip, extending counterclockwise from the noon side to the dusk side. This magnetic field configuration may result from the complexity of Jupiter's rapidly rotating magnetic field structure[2,3] and its interaction with the $B_y$-dominated solar wind near 5 AU[59,60]. During the two events discussed in this study, the spacecraft was situated in the high latitudes of the southern hemisphere on the dusk side, as indicated in locations "1" and "2" in Fig. 3b, providing a unique opportunity to potentially detect the cusp. Furthermore, due to the rotation of Jupiter and the tilt angle of about 9.5 degrees between the rotation axis and the magnetic axis, the trajectory of Juno's footprint through the Zhang et al.[2] simulation under the magnetic axis would exhibit a circular shape in the polar region near the dusk side. The differences between the JSS coordinate system used in the observation and the magnetic coordinate system used in the sketch of Fig. 3c are detailed in Supplementary Note 1. The trajectory of the spacecraft footprint would cross the open magnetic field region near the dusk side at the particular time when the cusp is detected. The latitude of these distributions is higher than that of the main auroral oval.

## Discussion
Our findings detail the characteristics of Jupiter's cusp, contributing to the understanding of solar wind-magnetosphere interactions across celestial bodies. Although Jupiter's cusp was observed on the dusk side in contrast with Earth's noon-positioned cusp and Saturn's cusp observed biased towards the dayside, the particle properties within the cusp regions are similar. These analogous cusp characteristics suggest that similar microphysical processes govern the nature of cusps across different planets. But the distinct location of these

observations provides a better understanding of solar wind-magnetosphere coupling at rapidly rotating planets.

Recent studies using high-precision simulations[2,3] have provided evidence that Jupiter's magnetic field structure forms a distinctive spiral pattern that acts to inflate the polar and dawnside magnetosphere relative to dusk. This magnetic field configuration renders the location of Jupiter's magnetopause reconnection site sensitive to the east-west ($B_y$) component of the IMF, as illustrated in Fig. 4a, b. Both the IMF azimuthal angle[59] and the clock angle[60,61] around Jupiter are reported to be predominantly around ±90°, which suggests that the solar wind conditions around Jupiter are $B_y$-dominated. When the IMF exhibits an eastward direction, the magnetopause reconnection sites are situated on the north and southeast sides. Conversely, when the IMF is oriented westward, these reconnection sites occur on the south and northeast sides of Jupiter's magnetosphere[3]. In contrast, dayside magnetopause reconnection at Earth is dominated by the north-south component of the IMF. Depending on whether the IMF $B_z$ component is positive or negative, terrestrial magnetopause reconnection occurs either in the high-latitude lobe region or at the sub-solar point. A more detailed discussion of the magnetic reconnection pictures related to the Jovian cusp can be found in Supplementary Note 5.

Considering the spacecraft's direction of motion, alongside the velocity filtering effect of detecting dispersed ions on magnetic field lines moving in different convective directions, we can anticipate the ion dispersion patterns under eastward or westward IMF conditions, as depicted in Fig. 4c, d. The spacecraft is predicted to detect reversed ion dispersion for eastward IMF, whereas, for westward IMF, a normal ion dispersion is expected. Comparing with our data in Figs. 1 and 2, we suggest that the IMF was directed westward during Case 1 and

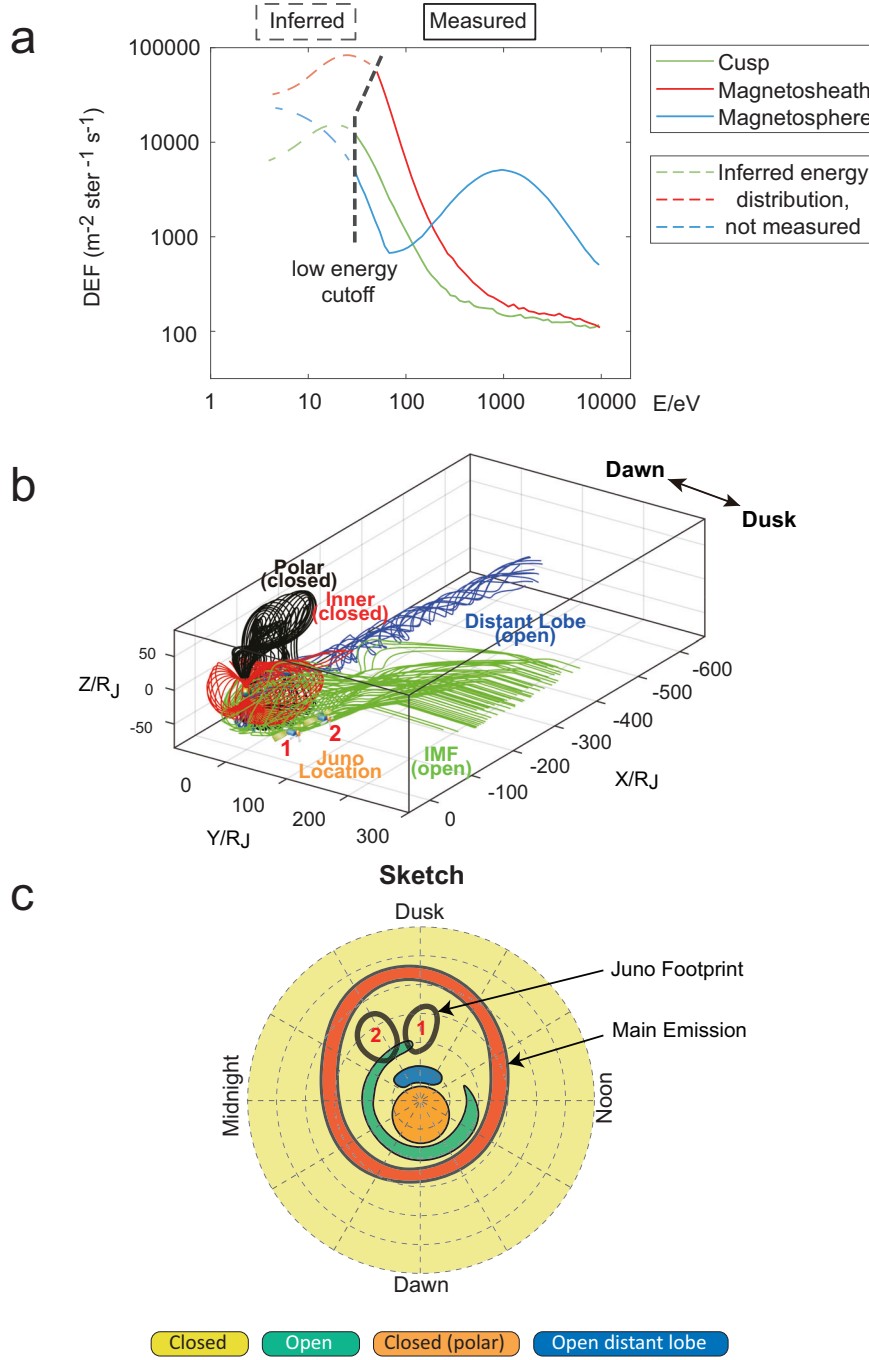

**Fig. 3 | The electron energy distributions and magnetic configuration associated with the Jovian cusp. a** The solid lines represent average electron energy distributions of the cusp Case 1 at 20:22 on 27 June – 02:12 on 28 June 2023 (green), magnetosphere on 27 June 2023 1501–1913 UT (blue), and magnetosheath (red) taken from the earlier study example on 2 October 2017 0900–1300 UT[49]. Noted that the dashed lines are inferred following the observational model[67], as direct measurement of electrons with energies below 50 eV would involve large uncertainties. The comparison between the three populations at Jupiter is similar to the results at Saturn[9]. **b** Jupiter's global magnetospheric topology and open-field configuration near dusk side based on the Zhang et al.[2] simulation. We suggest that Juno is in the position indicated. The labels "1" and "2" in panel (**b**) indicate the spacecraft positions during Case 1 and Case 2, respectively, in this study. **c** The diagram of the footprint distribution corresponding to the open and closed magnetic field regions in magnetic coordinates in the southern hemisphere based on the picture proposed by Zhang et al.[2]. The labels "1" and "2" in panel (**c**) indicate the spacecraft footpoint positions during Case 1 and Case 2, respectively, in this study.

eastward during Case 2. A more comprehensive analysis of the velocity filtering effect is provided in Supplementary Note 6. And see Supplementary Note 7 for a more detailed discussion on the comparison between the cusp and similar boundary layers.

In summary, our findings have provided useful insights into the nature of Jupiter's magnetospheric cusp. They demonstrate the consequences of rapid rotation on the location of reconnection sites and

Dungey-cycle driving of Jupiter's magnetosphere. The observational evidence presented here will contribute to comprehending the intricate interactions between the solar wind and Jupiter's magnetosphere. This work will help form a more comprehensive picture of space weather, particularly its role in the physical connections between stars and planets—an asset to the exploration and knowledge of our solar system.

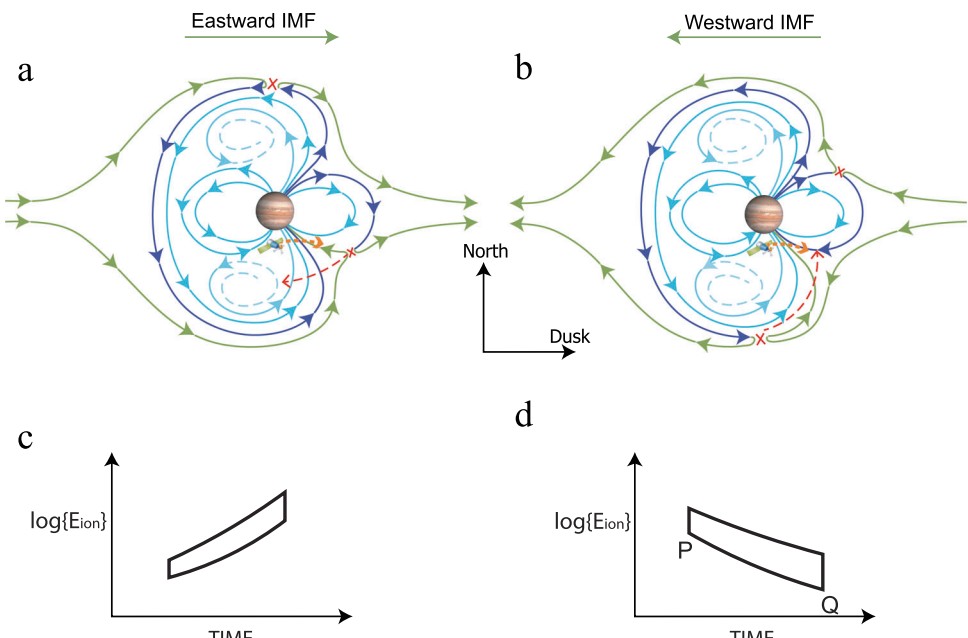

**Fig. 4 | The magnetopause reconnection picture at Jupiter and the predicted ion dispersion.** The distribution of reconnection sites of Jupiter's magnetic field configurations under **a** eastward and **b** westward solar wind conditions as revealed by high-precision simulations[2,3] as well as the schematic representation of the convective motion (red dashed lines) of the reconnected magnetic field line and the motion of the spacecraft in the sun view (orange dashed line); **c** The expected reversed ion dispersion under eastward solar wind conditions in this study; **d** The expected ion dispersion under westward solar wind conditions in this study. Green lines and dark blue lines are newly connected IMF and planetary field lines, respectively, that have reconnected at the reconnection sites identified by the red crosses. Light blue lines are closed, with the dashed portions in the north and south connecting to each other in the far tail.

## Methods

### Location of magnetic reconnection

The gradient of the ion dispersion is dependent on the distance to the reconnection site. For the case in this study, Eq. (1) can be used to roughly estimate the distance of proton flow from the reconnection site to the observation site[62]

$$E(\alpha_o, t) = \frac{M}{2t^2}\left[\int_{S_i}^{S_o}\frac{ds}{\sqrt{1 - \sin^2\alpha_o(B(s)/B_0)}}\right]^2 \tag{1}$$

where $E$ is the ion energy; $ds$ is arc length along a field line; $S_i$, $S_o$ are the observation and injection points; $M$ is the particle mass; $B(s)$ is the magnetic induction along the field line; $B_0$ is the magnetic induction at the observation point; $\alpha_o$ is the observed pitch angle; and $t$ is the transit time. Consider the simplest and farthest case of particle flow, i.e., the ion pitch angle is 0. Then we have Eq. (2)

$$E_P = \frac{M}{2t^2}\left[\int_I^P ds\right]^2 \tag{2}$$

where I refers to the injection point and A represents the observation point. Two representative points on the dispersion lowest edge, namely points P and Q, are selected for the calculation[29,63], as illustrated in Figs. 1d and 4d. As Fig. 1d shows, the time at point P is 20:35 on June 27, 2023 and the $E_P$ corresponds to 361.2 eV; The time recorded at point Q is 22:05, and the $E_Q$ corresponds to a value of 47.3 eV. Equation (2) outlines the formula for the motion of a proton from the injection point to point P, with the curve integral representing the distance the proton travels along the magnetic field line from the injection point to the observation point. Similarly, the equation showing the motion of the proton from the injection point to point Q

must also be taken into account, which is

$$E_Q = \frac{M}{2(t+T)^2}\left[\int_I^Q ds\right]^2 \tag{3}$$

where T is the interval time from point P to point Q. Since the distance from point P to point Q is very close and it is uncertain whether the spacecraft is traveling along the magnetic field lines, we can consider this factor to be 0, i.e., $\int_P^Q ds = 0$. So:

$$\int_I^Q ds \approx \int_I^P ds \tag{4}$$

Using Eqs. (2), (3), and (4), which are binary quadratic equations, we can obtain the proton's approximate travel distance from the injection point to observation point P:

$$\int_I^P ds \approx 8.1 R_J \tag{5}$$

Therefore, we can roughly surmise that Juno was approximately 8.1 $R_J$ along the magnetic field from the reconnection site. Similarly, we employ Eqs. (2) to (4) and the other two sets of dispersion edge points in Case 1 (the other two dispersion shown in Fig. 1d) to calculate the path distance from the injection point to the observation, obtaining approximately 7.7 $R_J$ and 5.6 $R_J$.

### Simulation information

The simulation results presented in this study are derived from Zhang et al.'s work[2] using the Grid Agnostic MHD (magnetohydrodynamic) for Extended Research Applications (GAMERA) global model[64]. In the solar-magnetospheric (SM) coordinate system, where the axes are

defined as X (Jupiter-Sun direction), Y (eastward/dusk), and Z (northward), GAMERA utilizes a specialized curvilinear, non-orthogonal grid to solve ideal MHD equations using a finite-volume method. The computational model employs a grid of $256 \times 256 \times 256$ cells[2], stretching 1200 $R_J$ along the X-axis and 400 $R_J$ perpendicular to it. Excluded from the computational domain is a spherical region with a radius of 6 $R_J$, centered on Jupiter and extending 100 $R_J$ downstream from the point where the solar wind and IMF conditions are imposed. To represent ion mass loading from the Io plasma torus, the model uses a spatial function with a fixed ion mass loading rate of ~1000 kg per second, focused at 6 $R_J$ in the equatorial plane[2]. Reflecting typical solar wind conditions[65,66], the upstream IMF $B_y$, solar wind density, dynamic pressure, and velocity are set at 0.5 nT, 0.2 cm$^{-3}$, 0.03 nPa, and 400 km s$^{-1}$, respectively.

## Data availability

All Juno data presented here are publicly available from NASA's Planetary Data System (https://pds-ppi.igpp.ucla.edu/). The MAG dataset is available via https://pds-ppi.igpp.ucla.edu/collection/JNO-J-3-FGM-CAL-V1.0, The JADE data is available via https://pds-ppi.igpp.ucla.edu/collection/JNO-J_SW-JAD-3-CALIBRATED-V1.0, and Wave dataset is available via https://pds-ppi.igpp.ucla.edu/collection/JNO-E_J_SS-WAV-3-CDR-SRVFULL-V2.0. The simulation data of Jupiter's global magnetospheric topology presented in this paper are publicly available online via https://doi.org/10.17605/OSF.IO/38WFE. The datasets generated during and/or analysed during the current study are available from the corresponding author upon request.

## Code availability

The Juno data were processed and analyzed using the Spedas package[68], which can be downloaded via the https://github.com/spedas page. The presentation of simulated data can be done by any standard plotting codes, for example, the Python function matplotlib.integrate (which can be downloaded via the https://matplotlib.org/stable/install/ page).

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

## Acknowledgements

Z.Y. was supported by the National Natural Science Foundation of China (No. 42388101). B.Z. is supported by the Excellent Young Scientists Fund (HongKong and Macau) of the National Natural Science Foundation of China (Grant No.41922060), the General Program of the National Natural Science Foundation of China (Grant 42374216), and the Research Grants Council (RGC) General Research Fund (Grant Nos.17308221, 17308520, 17315222, and 17308723). S.V.B. was supported by STFC grant ST/V000748/1. W.R.D. is supported by the Ernest Rutherford Fellowship: ST/W003449/1. F.A. and R.W.E. are funded through the Juno mission. The research at the University of Iowa is supported by NASA through Contract 699041X with the Southwest Research Institute. W.S.K. acknowledges the use of the Space Physics Data Repository at the University of Iowa supported by the Roy J.Carver Charitable Trust.

## Author contributions

Z.Y. proposed the original framework for the work. Y.X. conceived the specific research methodologies, refined the initial framework, carried out data analyses, and wrote the initial draft. C.S.A. created Fig. 3c and provided overall guidance for data analyses and research methodologies. B.Z. prepared the data of Fig. 3b and contributed images and tools for the representation of Jupiter's magnetospheric configuration. L.C.R. and S.V.B. provided guidance and assistance in identifying events and confirming magnetic reconnection pictures and data analyses. W.R.D. offered expertise in the field of Jupiter's magnetospheric dynamics and provided significant suggestions on improving the expression of the manuscript. R.W.E. contributed case studies on boundary layers, provided boundary layer cases to contrast with cusp regions, and participated in improving the manuscript and figures. J.J.C. helped in understanding the picture of magnetic reconnection related to Jupiter's cusp, providing figures of Jupiter's simulated magnetic configuration. F.A., W.S.K., J.E.P.C., and S.J.B. provided magnetic field data from Juno, as well as data from the JADE and Wave instruments. Y.X., Z.Y., C.S.A., L.C.R., S.V.B., W.R.D., F.A., W.S.K., D.J.M., Y.W., and T.S.Q. contributed to manuscript revisions and interpretation of results.

## Competing interests

The authors declare no competing interests.
