## [Peer Review File · Nature Communications]

Editorial Note: Parts of this peer review file have been redacted as indicated to avoid any copy right infringement.REVIEWER COMMENTS

Reviewer #1 (Remarks to the Author):

I am sorry for the delay in reviewing Nature Communications paper NCOMMS-23-49912-T. Although I find the paper interesting and think there is merit in what the authors are attempting to do, I have been struggling with the approach used in this paper. As presented the paper is data poor and does not provide quantitative criteria to identify a cusp at Jupiter - if it was indeed observed.

The claim of a cusp at Jupiter based on a Juno data set that contains only one example is far from compelling. Although this paper seems to provide an example of reconnection at Jupiter, the nature and characteristics of this event are not sufficient to claim that one has identified a cusp region at Jupiter. Some of the ion and electron energy spectral characteristics seem compatible with cusp-like features, but the location of the event is unexpected. The authors seem to circumvent this lack of compelling evidence by referring to the published magnetospheric interaction simulation of Zhang et al and showing compatibility between the Juno 'cusp location' and the simulation results. This would appear to me to be more of a pivot than solid evidence that the Juno data set is indicative of the cusp.

At present the most compelling evidence presented in the paper is the energy dispersion signatures that are shown, yet there are numerous examples of ion energy dispersion in Jupiter's magnetosphere most of which have nothing to do with the existence of a cusp.

Alternatively, if the purpose of the paper is to show that the Juno data can be used to validate the Zhang et al simulation then this should be the expressed goal of the paper and not a claim of finding Jupiter's cusp region. In either case this reviewer would insist that several data examples of the phenomena and/or set of ion energy dispersion examples establishing the uniqueness of this proposed cusp example must be presented and a very consistent pattern identified that illustrates the existence of a cusp. If on the other hand, the goal of the paper is to validate the Zhang et al simulations then a series of data set examples of Juno observations must be presented that span the simulation regions of note and that are used to build a compelling case that the Zhang et al simulations represent reality.

Furthermore, no clear definition and comparison of a cusp at Earth, Jupiter, or Saturn are introduced in a manner that allows a compelling metric to be developed which would verify the present conjecture. Given the content of the present paper, the authors have simply observed a likely reconnection event in a different place in Jupiter's magnetosphere, but have failed to convince this

reviewer that this is a cusp (or to validate the Zhang et al. simulation) based on the existence of one data set.

Reviewer #2 (Remarks to the Author):

Xu et al., present Juno particle observations from high latitude dusk observations at Jupiter's magnetosphere which they identify as being cusp plasma, i.e. plasma from the shocked solar wind which has been injected upon the reconnection of a closed magnetospheric field with the interplanetary magnetic field. The key identifying feature which the authors use to identify the cusp plasma are plasma energies similar to the magnetosheath. If this observation was indeed of the cusp, then this would provide a significant contribution to our knowledge of the Jovian magnetosphere.

The high energy cutoff labelled A and B are irrelevant to the analysis of the cusp dispersion (i.e., Figure 3 e and 4 d). The focus should be on the low-energy cutoff of the ions (i.e. C and whatever the lowest energy is below 'A'). The lowest energy ions observed at any particular time are expected to be from the reconnection site, since higher energy ions can enter from the magnetosheath along the open field line after reconnection has occurred and the field line has convected and be measured at the same time as the lowest energy. This will change your estimated energies which will affect your distance calculation.

The authors use, what is assumed to be a magnetic field model by Zhang et al., (2021) to show schematics of a map of the magnetosphere and the location of Juno in the polar ionosphere. This is used as a way of attempting to persuade the reader that the spacecraft crossed through the open field region. Figure 2 shows that during the data timeseries, Juno does not significantly change location in Local time or Latitude. However, according to Figure 1c Juno travels towards the dayside and then goes to the nightside again, which is confusing.

Response to Referee #1's comments

We would like to extend our sincere appreciation to the referee for their valuable suggestions and insightful comments. Below are our point-by-point responses to the referee's remarks. The original comments of the referee are quoted *in italics* for convenience. Corresponding revisions have been made according to the referee's suggestions. The line numbers in our response refer to the highlighted draft.

Reviewer #1 (Remarks to the Author):

I am sorry for the delay in reviewing Nature Communications paper NCOMMS-23-49912-T. Although I find the paper interesting and think there is merit in what the authors are attempting to do, I have been struggling with the approach used in this paper. As presented the paper is data poor and does not provide quantitative criteria to identify a cusp at Jupiter - if it was indeed observed.

Response #1

We deeply appreciate the referee for providing the valuable insights and constructive feedback on our manuscript. Following the referee's major concern about the evidence presented for identifying a cusp region at Jupiter, we have now significantly improved the manuscript in the following three points:

- (1) In the revised manuscript, we now make a comparison of identification criteria for Earth, Saturn, and Jupiter's cusps. We believe a global comparison on cusps at the three planets would provide a clear picture of planetary cusp (or cusp-like structures).
- (2) A more quantitative analysis is performed, such as the comparison of electron distributions in the magnetosphere, magnetosheath and cusp.
- (3) Based on a set of key characteristics of cusp, we have surveyed all available data from Juno, and identified several more events from the most recent observations (i.e., in 2023). The consistent observations from previously predicted region further support the magnetospheric picture as revealed by numerical simulation.

We believe that the improved manuscript, with additional events and quantitative analysis, can now provide solid support to the identification of Jovian cusp. Again, thanks for the constructive comments/suggestions. The one-to-one responses to the detailed comments are listed below.

The claim of a cusp at Jupiter based on a Juno data set that contains only one example is far from compelling. Although this paper seems to provide an example of reconnection at Jupiter, the nature and characteristics of this event are not sufficient to claim that one has identified a cusp region at Jupiter. Some of the ion and electron energy spectral characteristics seem compatible with cusp-like features, but the location of the event is unexpected. The authors seem to circumvent this lack of compelling evidence by referring to the published magnetospheric interaction simulation of Zhang et al and showing compatibility between the Juno 'cusp location' and the simulation results. This would appear to me to be more of a pivot than solid evidence that the Juno data set is indicative of the cusp. At present the most compelling evidence presented in the paper is the energy dispersion signatures that are shown,

yet there are numerous examples of ion energy dispersion in Jupiter's magnetosphere most of which have nothing to do with the existence of a cusp.

Response #2

The key goal of this study is to report the discovery of Jupiter's cusp region using observations from Juno in high latitude region. The re-analysis of the global MHD simulations in Zhang et al. (2021) and Chen et al. (2023) provide additional support in understanding Jupiter's global magnetic field configuration, which helps the interpretation of the observed cusp in an unexpected region. The magnetosheath-like plasma population in relative high latitude well inside the magnetopause is the key feature of the cusp in terrestrial and Kronian magnetospheres. The energy dispersion is additional supporting evidence associated with boundary dynamics. In previous manuscript, we only identified one event, which indeed would bring the concern of other potential explanation.

Following the suggestions by the referee, we have made several improvements to strengthen the identification and presentation of the cusp structure. These improvements are categorized into two primary **sections (1 and 2)**:

(1) We clarified and compared the identification criteria for cusps on Earth, Saturn, and Jupiter. Now we established clear and robust identification criteria specifically for Jupiter's cusp following the well-established terrestrial cusp picture, including the magnetic or plasma features (e.g., ion energy dispersion etc).

(2) We surveyed all Juno datasets to date, and have identified more cusp cases. The manuscript has been improved to incorporate additional typical cusp events. With the study of 5 more typical example events including more than 10 cusp crossings, the identification of Jupiter's cusp is much more robust, and the cusp features are presented as a consistent picture that provides reliable guidance to future investigations.

Regarding the referee's concern about the unexpected location of the dusk-side cusp, in **section (3)**, we provide a possible explanation in detail by using the high-resolution simulation from Zhang et al. (2021) and Chen et al. (2023).

The following is the detailed description about the three sections.

(1) Criteria for the Identification of Earth's, Saturn's and Jupiter's Cusps.

As shown in **Table R1**, based on previous studies, the identification criteria for the cusps of Earth, Saturn, and Jupiter (inferred) are categorized into three levels of importance: Level 1, Level 2, and Level 3, in descending order. The Level 2 and 3 would more depend on the spacecraft location, solar wind conditions and so on, e.g., spacecraft location relative to cusp-associated reconnections, while the Level 1 features are mandatory for the identification. Each criterion is elucidated and discussed sequentially below.

Importance and Reliability: Level 1 > 2 > 3

Identification of Cusp	Level 1 (key)				Level 2		Level 3		
	Inside Magnetopause	High Latitude Location	Magnetosheath-like Plasma/Electron	Magnetic Depression	Reconnection-Related Ion Dispersion	-	Whistler-mode Hiss Wave	Electron Pitch Angle Features	Other
Earth	Inside Magnetopause	High Latitude Location	Magnetosheath-like Plasma/Electron	Magnetic Depression	Reconnection-Related Ion Dispersion	-	Whistler-mode Hiss Wave	Electron Pitch Angle Features	Other
Saturn	Inside Magnetopause	High Latitude Location	Magnetosheath-like Plasma/Electron	-	Reconnection-Related Ion Dispersion	Magnetic Depression	Whistler-mode Hiss Wave	Electron Pitch Angle Features	Other
Jupiter (inferred)	Inside Magnetopause	High Latitude Location	Magnetosheath-like Plasma/Electron	-	Reconnection-Related Ion Dispersion	Magnetic Depression	Whistler-mode Hiss Wave	Electron Pitch Angle Features	Other

Table R1. Summary comparison of cusp identification criteria for Earth, Saturn and inferences of identification criteria for Jupiter's cusp.

- Level 1 identification criteria for cusp (most important).

- (a) Magnetosheath plasma features at high latitude inside the magnetosphere.

The magnetospheric cusp of a planet can be defined as part of the magnetosphere in the vicinity of the polar region at high magnetic latitudes/ invariant latitude, where a significant quantity of magnetosheath plasma is detected inside the magnetopause position (e.g., Cargill et al., 2005; Spreiter et al., 1966a; Smith and Lockwood et al., 1996). Defined by its nature and position, a cusp is a region where magnetosheath plasma and momentum can enter the magnetosphere. Therefore, the spacecraft's high-latitude positioning and the detection of magnetosheath plasma constitute the primary criteria for cusp identification., high-latitude location and magnetosheath plasma/electron features are listed as 'level 1' identification criteria.

At Earth, magnetosheath particles entering the cusp typically include low energy ions within the 500 eV–5 keV range and soft electrons around 50 eV (e.g., Lin et al., 1986; Pitout & Bogdanova, 2021). Low-energy magnetosheath-like electron distributions (10s ~ 100s eV enhancement) have been used to identify cusp events (in conjunction with the spacecraft's inside-magnetopause and high-latitude position) since in-situ observations in 1971 (Heikkila & Winningham, 1971; Frank, 1971), and this identification criterion has been used for decades (e.g., Lin et al., 1986; Lockwood et al., 1993; Cargill et al. 2005). The distinction between Earth's cusp and other boundary layers has utilized the criterion of average electron spectrum enhancement, with levels $E_e < 220$ eV indicating magnetosheath-like conditions typical for the cusp, and enhancements from $220 < E_e < 600$ eV identifying other boundaries (e.g., cleft/LLBL, Newell et al., 1988, 1989). Thus, magnetosheath-like electrons at high latitudes within the magnetosphere constitute a dependable criterion for identifying cusp regions, a methodology that has also been applied to the identification of Saturn's cusp (~100 eV electron enhancement, Jasinski et al., 2014, 2016; Arridge et al., 2016). In contrast, sheath-like ion features are more variable and less reliable for identification purposes than electrons, due to the velocity filtering effects linked with reconnection and the intermittent nature of cusp-related reconnection (e.g., Lavraud et al., 2004). While electrons are also accelerated during reconnection, the velocity change is negligible (e.g., Smith & Lockwood, 1996).

- (b) Magnetic depression.

It should be noted that magnetic depression features are also important for identifying cusp at Earth (e.g., Tsyganenko & Russell 1999; Dunlop et al. 2005; Zhang et al. 2013). The magnetic depression observed in the cusp region results from elevated plasma pressure and density due to the inflow of magnetosheath hot plasma, which is a magnetic turbulence feature that accompanies the magnetosheath inflow. In the cusp region, plasma thermal and magnetic pressures maintain a state of relative equilibrium. An increase in plasma thermal pressure leads to a significant reduction in magnetic pressure within the outer cusp, ensuring the maintenance of pressure equilibrium with the adjacent magnetospheric regions (e.g., Zhou et al., 2001; Shi et al., 2009). Lavraud et al. (2004) statistically demonstrated the presence of a diamagnetic cavity within the cusp region.

However, magnetic depression features are not always present in Saturn cusp events. Magnetic depression features are inconsistently observed in Saturn's cusp events. While some cases exhibit marked magnetic depressions akin to Earth's, others display weak or no magnetic field disturbances (e.g., see Figure 4, overview of Jan 2007, Arridge et al., 2016). Jasinski et al. (2017) reported that only two out of five winter cusp events exhibited magnetic depressions. Similarly, analysis of Juno data indicates that Jupiter's cusp events, as displayed in section (2), largely lack magnetic depression features. The absence of this feature for some cusp cases of giant planets remains unresolved. A plausible explanation considers the correlation between the plasma thermal pressure in the cusp and the solar wind dynamic pressure surrounding the planet. The solar wind dynamic pressure at the orbits of the giant planets (Jupiter at 5 AU, Saturn at 9 AU) is significantly lower than at Earth, while their magnetic field strengths in the cusp regions are comparable (10-100 nT). Consequently, the plasma from the magnetosheaths of these giant planets enters the cusp with generally lower thermal pressure relative to magnetic pressure, not necessarily inducing magnetic depression. This relationship is attributed to the solar wind's thermal pressure being minor compared to its ram pressure, with the latter being a major contributor to magnetosheath pressure (Lavraud et al., 2004), which, in turn, influences the thermal pressure of plasma entering the cusp. According to Spreiter et al. (1966b), magnetosheath pressure, a linear function of the solar wind Mach number squared (M^2), can be approximated as proportional to V_{sw}^2 , and magnetosheath density correlates with solar wind density. Therefore, within reasonable Mach number ranges, magnetosheath pressure correlates with ρV_{sw}^2 . Lavraud et al. (2004) conducted a statistical analysis on the ratio of plasma thermal pressure to solar wind dynamic pressure inside and outside the cusp, finding the magnitudes to be very comparable (see Figure 6b in it). Furthermore, Zhou et al. (2001) and Jasinski et al. (2017) statistically examined the correlation between cusp magnetic depression and solar wind dynamic pressure for Earth and Saturn, respectively, with the latter utilizing the Michigan Solar Wind Model. Their findings indicated that higher solar wind dynamic pressures lead to more pronounced cusp magnetic depressions. And studies on Mercury (Winslow et al., 2012; Raines et al., 2014; Slavin et al., 2014; Poh et al., 2016) have shown magnetic depression to be stronger than at Earth. These investigations collectively highlight the significant influence of the solar wind environment on cusp magnetic depression phenomena.

In summary, based on the cusp's definition and extensive observational reports, the presence of magnetosheath-like electron features at high latitudes well within the magnetopause is determined as the paramount criterion for identifying Jupiter's cusp. Magnetic depression, considered a key identification criterion for Earth, is not deemed a necessary feature for the giant planets. Therefore, it is classified as a Level 2 criterion.

- Level 2 identification criteria for cusp (less important but helpful).

The cusp, being the region where magnetosheath plasma and momentum enter the magnetosphere, exhibits plasma characteristics that are significantly influenced by magnetic reconnection. At Earth, the cusp's ion characteristics are affected by the magnetic reconnection process occurring at low latitude (Reiff et al., 1977; Maynard et al. 1991) or in high-latitude lobe region (Øieroset et al. 1997; Bosqued et al., 1985). Numerous studies have shown that, within Earth's cusp, ions often display dispersion (e.g., Reiff et al., 1977; Maynard et al. 1991; Pitout et al. 2009) or "reversed" dispersion patterns (e.g., Woch & Lundin, 1992; Øieroset et al. 1997), which have been used as an important feature in identify cusp. These patterns arise from the velocity filtering effect linked to acceleration during magnetic reconnection. The $E \times B$ drift causes high-energy ions to reach a given altitude before their slower counterparts, which arrive at the same altitude but at a different location, such as latitude. This time-of-flight effect, in combination with transverse convection, results in the well-known ion dispersion (e.g., Reiff et al., 1997; Lockwood & Smith, 1994). Different dispersion characteristics correspond to different solar wind conditions and reconnection locations (e.g., Woch & Lundin, 1992; Pitout et al. 2009). Normal and reversed ion dispersion features have also been used to identify Saturn's cusp (Jasinski et al., 2014; Jasinski et al., 2016; Arridge et al., 2016).

However, at Earth, due to the cusp's dynamic nature and the sporadic occurrence of magnetic reconnection, plasma flows within the cusp often exhibit intermittent features (e.g., Haerendel et al., 1978; Bosqued et al., 2005; Escoubet et al., 2006; Pitout & Bogdanova, 2021), which are affected by the solar wind and the spacecraft's position. Consequently, not all observations within the cusp show clear ion dispersion. Reports on Saturn's cusp indicate similar findings (Jasinski et al., 2016; Arridge et al., 2016). Thus, ion dispersion is classified as a Level 2 criterion, underlining its significance yet acknowledging it is not necessity for cusp identification. Furthermore, as outlined in section (b) regarding Level 1 criteria, magnetic depressions are also categorized under Level 2.

- Level 3 identification criteria for cusp (not necessary but can help).

At Earth, enhanced plasma waves ranging from 1 to 100s Hz—where the upper limit is the electron cyclotron frequency—are often utilized in conjunction with magnetic fields and plasma flows to identify the cusp (e.g., Chen et al., 1997, 1998; Fung et al., 1997). These auroral-like hiss waves are believed to consist of whistler mode emissions (Gurnett & Frank, 1978; Chen et al., 1998) and are thought to be produced by a Cerenkov radiation mechanism (Maggs, 1976). Similarly, auroral hiss has been detected in the very high-latitude magnetosphere of Jupiter (Gurnett et al., 1979), akin to the hiss observed over

Earth's auroral zones . It has been proposed that these emissions were generated by field-aligned electron beams (Gurnett et al., 1979), which may arise from reconnection-related processes. Stone et al. (1992) considered observations of enhanced auroral hiss near 100 Hz as potential indicators of Jupiter's cusp. Although less extensively studied at Saturn, it has also shown associations between auroral hiss and cusp regions. Auroral hiss Enhancements around 100 Hz have been observed within parts of the cusp, and in adjacent areas (Jasinski et al., 2014; Arridge et al., 2016).

In summary, auroral hiss can be used to assist in the identification of cusp, which can be generated by field-aligned electron beams produced by cusp-associated reconnection processes. However due to the sporadic nature of reconnection and the complexity of the cusp environment, auroral hiss is not consistently present in cusp regions, and thus it is classified as a Level 3 criterion, which can help identify cusp but is not necessary.

Similarly, field-aligned electron pitch angle features (enhancements near 0 and 180°), linked to reconnection processes, can facilitate the identification of the cusp on both Earth and Saturn (Smith & Lockwood, 1996; Jasinski et al., 2014; Arridge et al., 2016). However, there may be other properties of the electron pitch angle in the cusp, for example, enhancements around 90° due to local acceleration related to gradients in reconnected quasi-potential (Nykyri et al., 2012) and wave-particle interactions (Nykyri et al., 2004; Grison et al., 2005). Consequently, given the variable nature of electron pitch angle data within the cusp, this feature is also classified as a Level 3 criterion—helpful for identification but not very important.

(2) Presentation of Features of all Cusp Cases.

This section summaries the features of all cusp cases in this study (Figure R1-R6), all well meeting the Level 1 criteria for Jupiter. Moreover, the majority of these events also exhibit characteristics outlined in Levels 2 and 3 as listed in Table R1 (e.g., ion dispersion and hiss wave), with the notable exception of magnetic depression. Another possible explanation for the lack of magnetic depressions is the local time all cases occur close to the dusk side, away from local noon. Statistical studies of the Earth's cusp have demonstrated that the closer the local time is to noon, the more pronounced the magnetic depression (Zhou et al., 2001). Similarly, cusp events at Saturn showing magnetic depression features were observed near noon (Arridge et al., 2016; Jasinski et al., 2017).

Identification of Cusp	Level 1 (key)			Level 2		Level 3		
	Inside Magnetopause	High Latitude Location	Magnetosheath-like Plasma/Electron	Reconnection-Related Ion Dispersion	Magnetic Depression	Whistler-mode Hiss Wave	Electron Pitch Angle Features	Other
Jupiter (inferred)								
Jovian cusp case 1	√	√	√	√	-	√	-	-
Jovian cusp case 2	√	√	√	√	-	√	√	-
Jovian cusp case 3	√	√	√	√	-	√	-	-
Jovian cusp case 4	√	√	√	√	-	√	-	-
Jovian cusp case 5	√	√	√	√	-	√	-	-
Jovian cusp case 6	√	√	√	√	-	√	√	-

Table R2. The list which outlines how the criteria are met for all cusp cases. Due to the poor coverage of electron pitch angle data across most event observations, features such as field-aligned electrons are not discernible in the majority of events.

- Typical ion energy dispersion

All cusp cases in this study all well meet the Level 1 and ion dispersion (normal or reversed) Level 2 criteria for Jupiter. Furthermore, all events also exhibit enhanced auroral hiss characteristics outlined in Level 3 as listed in **Table R1**. The identification of field-aligned electron features is challenging due to the absence of significant electron data near the pitch angles of 0 and 180 degrees in most events. The compliance of all events with the identification criteria is detailed in **Table R2**.

Two cases shown in **Figure R1** and **R2**, exemplifying typical ion energy dispersion and reversed ion energy dispersion, are elaborately discussed in the main text. The case in the original manuscript was replaced by **Figure R1** and **R2** due to its short duration (only half an hour) and less clear dispersion features. For the analysis of each event, refer to the revised manuscript and supplementary materials. Each case shown contains three parts, the case data, Juno's footpoint distribution, and Juno's location.

Figure R1. Cusp example 1 showing clear typical ion dispersion (panel d). (a) R-Theta-Phi magnetic field components in JSS (Jupiter-De-Spun-Sun) coordinate; (b) The total magnetic field strength; (c) The electron energy spectrogram; Ion Energy spectrogram for protons (d) and heavy ions (e), where heavy ions represent ions with m/q in the range of 5 and 6428; (f) Pitch angle distribution for electrons which is normalized at each time unit within energy ranges of

0.3 to 32 keV; (g) Plasma wave observations in the frequency range 50 to 300 Hz. The different regions that the spacecraft passes through are marked with different colors at the top and separated by dashed lines. 'M' is the magnetosphere, 'C' is the cusp, 'BL' is the boundary layern. The red arrows and white dashed line in panel (d) show the dispersion. The yellow arrows in panel (g) indicate the enhanced auroral hiss features. The blue dashed line demarcates the cusp into two regions, labeled as 'a' and 'b', each characterized by different plasma properties. (h) Traced distribution of spacecraft footprints before and after cusp observation in Left-Handed system III coordinates. The red regions are the main ovals, and the colored lines are the Juno footprint trajectories before and after the cusp region observation (June 26th to April 28th, 2023). (i) The position of the spacecraft around cusp observations, red lines representing the time range of the cusp case.

Figure R2. Cusp example 2 showing clear reversed ion dispersion (panel d), using the same format as Figure R1.

Figure R3. Cusp example 3 showing clear normal ion dispersion (panel d), using the same format as Figure R1.

Figure R4. Example 4 of Cusp observations, using the same format as Figure R1.

Figure R5. Example 5 of cusp observations, using the same format as Figure R1.

Figure R6. Example 6 of cusp observations, using the same format as Figure R1, which is the event in the original manuscript.

(3) The Understanding of the Unexpected Duskside Location of Cusp.

The auroral morphologies observed on Jupiter, differing from those on Earth and Saturn, suggest an unusual magnetic field configuration and cusp distribution that sets Jupiter apart from the traditional auroral pictures associated with Earth. Terrestrial auroral emissions are generated by the precipitation of energetic particles along magnetic field lines, as shown in Figure R7a (Eather & Mende, 1971; Eather, 1967). These emissions form an auroral oval that encircles the magnetic pole (Figure 1a). In contrast, although Saturn's magnetic fields are also predominantly dipolar, its auroral emissions display significant variations with complex morphologies (G rard et al., 2006; Grodent et al., 2005; Carbary, 2012), exhibiting features characteristic of both Earth-like and Jupiter-like structures (Figures R7b-d). At Jupiter, the aurora present a distinctly different morphology and evolutionary pattern as depicted in Figure R7e (Clarke et al., 2012; Greathouse et al., 2021; Sulaiman et al., 2022), characterized by a highly dynamic yet generally bright polar cap aurora and contrasting dark "polar collar" regions.

Zhang et al. (2021) provide the only study to date that comprehensively explains these different auroral morphologies on Jupiter, whose insights into Jupiter's magnetic field structure have also established a crucial foundation for understanding the unusual cusp locations observed in this study. The formation and different configuration of Jupiter's cusp can potentially be attributed to the solar wind environment around Jupiter and the asymmetrical configuration of Jupiter's rapidly rotating magnetosphere. Shown by the simulation result in Chen et al. (2023), the strong centrifugal force exerted during Jupiter's rapid rotation causes its magnetospheric configuration to appear "flatter," resulting in a greater presence of topological eastward/westward components, as shown in Figure R8. This flattened structure is suggested to be more sensitive to east-west solar wind-induced magnetic reconnection, as shown in Figure R9.

Furthermore, previous research indicates that both the interplanetary magnetic field (IMF) azimuthal angle (Ebert et al., 2014) and the clock angle (Nichols et al., 2006, 2017) around Jupiter are predominantly around $\pm 90^\circ$, suggesting that the solar wind near Jupiter is primarily east-west oriented (B_y component dominant). Displayed by the simulation results in Chen et al. (2023), this orientation facilitates a unique coupling of the solar wind to Jupiter's magnetosphere, which exhibits a distinct magnetic field configuration, as depicted in Figure R9. This configuration allows for the existence of open magnetic field lines coupled to the solar wind on the dusk side inside the magnetosphere. It is within this portion of the magnetosphere that the cusp region can be observed, as shown in the boxes in Figures R9a and R9b.

Corresponding revisions and additions have been made in the revised manuscript, please see line 95 to 186, line 196 to 201, line 209 to 216, line 240 to 245 in the revised manuscript and Text S2, S3 and S5 in the Supplementary Information.

Figure R8. Comparison of (a) Earth's and (b) Jupiter's magnetic configuration using Chen et al. (2023) simulation results. The former exhibits dawn-dusk symmetry, whereas the latter demonstrates dawn-dusk asymmetry attributed to the effects of rapid rotation.

Figure R9. Jupiter's magnetic configuration under different typical (a) eastward / (b) westward solar wind conditions. (c) and (d) depict the zoomed-in magnetospheric structure at the location of the cusp events under different solar wind conditions, using Chen et al. (2023) simulation results. The yellow spheres, labeled "event 1" and "event 2," denote the positions of the spacecraft during the pre-dusk cusp case 1 (**Figure R1**) and the post-dusk cusp case 2 (**Figure R2**), which are discussed in detail in the revised manuscript.

Reference

- Arridge, C. S., Jasinski, J. M., Achilleos, N., Bogdanova, Y. V., Bunce, E. J., Cowley, S. W., ... & Krupp, N. (2016). Cassini observations of Saturn's southern polar cusp. *Journal of Geophysical Research: Space Physics*, 121(4), 3006-3030.
- Bosqued, J. M., Sauvaud, J. A., Reme, H., Crasnier, J., Galperin, Y. I., Kovrazhkin, R. A., & Gladyshev, V. A. (1985). Evidence for ion energy dispersion in the polar cusp related to a northward-directed IMF. *Advances in space research*, 5(4), 149-153.
- Bosqued, J. M., Escoubet, C. P., Frey, H. U., Dunlop, M., Berchem, J., Marchaudon, A., ... & Balogh, A. (2005). Multipoint observations of transient reconnection signatures in the cusp precipitation: A Cluster-IMAGE detailed case study. *Journal of Geophysical Research: Space Physics*, 110(A3).
- Carbary, J. F. (2012). The morphology of Saturn's ultraviolet aurora. *Journal of Geophysical Research: Space Physics*, 117(A6).
- Cargill, P. J., Lavraud, B., Owen, C. J., Grison, B., Dunlop, M. W., Cornilleau-Wehrin, N., ... & Nykyri, K. (2005). Cluster at the magnetospheric cusps. *Space science reviews*, 118, 321-366.
- Chen, S. H., Boardsen, S. A., Fung, S. F., Green, J. L., Kessel, R. L., Tan, L. C., ... & Craven, J. D. (1997). Exterior and interior polar cusps: Observations from Hawkeye. *Journal of Geophysical Research: Space Physics*, 102(A6), 11335-11347.
- Chen, J., Fritz, T. A., Sheldon, R. B., Spence, H. E., Spjeldvik, W. N., Fennell, J. F., ... & Gurnett, D. A. (1998). Cusp energetic particle events: Implications for a major acceleration region of the magnetosphere. *Journal of Geophysical Research: Space Physics*, 103(A1), 69-78.
- Chen, J., Zhang, B., Lin, D., Delamere, P. A., Yao, Z., Brambles, O., ... & Lyon, J. G. (2023). Prediction of axial asymmetry in Jovian magnetopause reconnection. *Geophysical Research Letters*, 50(9), e2022GL102577.
- Clarke, J. T. (2012). Auroral processes on Jupiter and Saturn. *Auroral phenomenology and magnetospheric processes: Earth and other planets*, 197, 113-122.
- Dunlop, M. W., Lavraud, B., Cargill, P., Taylor, M. G. G. T., Balogh, A., Réme, H., ... & Marchaudon, A. (2005). Cluster observations of the cusp: magnetic structure and dynamics. *Surveys in Geophysics*, 26, 5-55.
- Eather, R. H. (1967). Auroral proton precipitation and hydrogen emissions. *Reviews of Geophysics*, 5(3), 207-285.
- Eather, R. H., & Mende, S. B. (1971). Airborne observations of auroral precipitation patterns. *Journal of Geophysical Research*, 76(7), 1746-1755.
- Ebert, R. W., Bagenal, F., McComas, D. J., & Fowler, C. M. (2014). A survey of solar wind conditions at 5 AU: A tool for interpreting solar wind-magnetosphere interactions at

- Jupiter. *Frontiers in Astronomy and Space Sciences*, 1, 4.
- Escoubet, C. P., Bosqued, J. M., Berchem, J., Trattner, K. J., Taylor, M. G. G. T., Pitout, F., ... & Fazakerley, A. (2006). Temporal evolution of a staircase ion signature observed by Cluster in the mid-altitude polar cusp. *Geophysical research letters*, 33(7).
- Frank, L. A. (1971). Plasma in the earth's polar magnetosphere. *Journal of Geophysical Research*, 76(22), 5202-5219.
- Fung, S. F., Eastman, T. E., Boardsen, S. A., & Chen, S. H. (1997). High-altitude cusp positions sampled by the Hawkeye satellite. *Physics and Chemistry of the Earth*, 22(7-8), 653-662.
- Gérard, J. C., Grodent, D., Cowley, S. W. H., Mitchell, D. G., Kurth, W. S., Clarke, J. T., ... & Coates, A. J. (2006). Saturn's auroral morphology and activity during quiet magnetospheric conditions. *Journal of Geophysical Research: Space Physics*, 111(A12).
- Greathouse, T., Gladstone, R., Versteeg, M., Hue, V., Kammer, J., Giles, R., ... & Vogt, M. F. (2021). Local time dependence of Jupiter's polar auroral emissions observed by Juno UVS. *Journal of Geophysical Research: Planets*, 126(12), e2021JE006954.
- Grison, B., Sahraoui, F., Lavraud, B., Chust, T., Cornilleau-Wehrin, N., Reme, H., ... & André, M. (2005, December). Wave particle interactions in the high-altitude polar cusp: a Cluster case study. In *Annales Geophysicae* (Vol. 23, No. 12, pp. 3699-3713). Göttingen, Germany: Copernicus Publications.
- Grodent, D., Gérard, J. C., Cowley, S. W. H., Bunce, E. J., & Clarke, J. T. (2005). Variable morphology of Saturn's southern ultraviolet aurora. *Journal of Geophysical Research: Space Physics*, 110(A7).
- Gurnett, D. A., & Frank, L. A. (1978). Plasma waves in the polar cusp: Observations from Hawkeye 1. *Journal of Geophysical Research: Space Physics*, 83(A4), 1447-1462.
- Gurnett, D. A., Kurth, W. S., & Scarf, F. L. (1979). Auroral hiss observed near the Io plasma torus. *Nature*, 280(5725), 767-770.
- Haerendel, G., Paschmann, G., Sckopke, N., Rosenbauer, H., & Hedgecock, P. C. (1978). The frontside boundary layer of the magnetosphere and the problem of reconnection. *Journal of Geophysical Research: Space Physics*, 83(A7), 3195-3216.
- Heikkila, W. J., & Winningham, J. D. (1971). Penetration of magnetosheath plasma to low altitudes through the dayside magnetospheric cusps. *Journal of Geophysical Research*, 76(4), 883-891.
- Jasinski, J. M., Arridge, C. S., Lamy, L., Leisner, J. S., Thomsen, M. F., Mitchell, D. G., ... & Waite, J. H. (2014). Cusp observation at Saturn's high-latitude magnetosphere by the Cassini spacecraft. *Geophysical Research Letters*, 41(5), 1382-1388.
- Jasinski, J. M., Arridge, C. S., Coates, A. J., Jones, G. H., Sergis, N., Thomsen, M. F., ... & Waite Jr, J. H. (2016). Cassini plasma observations of Saturn's magnetospheric cusp. *Journal of Geophysical Research: Space Physics*, 121(12), 12-047.
- Lavraud, B., Fedorov, A., Budnik, E., Grigoriev, A., Cargill, P. J., Dunlop, M. W., ... & Balogh, A. (2004, September). Cluster survey of the high-altitude cusp properties: a three-year statistical study. In *Annales Geophysicae* (Vol. 22, No. 8, pp. 3009-3019). Göttingen, Germany: Copernicus Publications.
- Lin, C. S., Burch, J. L., & Winningham, J. D. (1986). Near-conjugate observations of polar cusp electron precipitation using DE 1 and DE 2. *Journal of Geophysical Research: Space Physics*, 91(A10), 11186-11202.

- Lockwood, M., Denig, W. F., Farmer, A. D., Davda, V. N., Cowley, S. W. H., & Lühr, H. (1993). Ionospheric signatures of pulsed reconnection at the Earth's magnetopause. *Nature*, 361(6411), 424-428.
- Lockwood, M., & Smith, M. F. (1994). Low and middle altitude cusp particle signatures for general magnetopause reconnection rate variations: 1. Theory. *Journal of Geophysical Research: Space Physics*, 99(A5), 8531-8553.
- Maggs, J. E. (1976). Coherent generation of VLF hiss. *Journal of Geophysical Research*, 81(10), 1707-1724.
- Maynard, N. C., Aggson, T. L., Basinska, E. M., Burke, W. J., Craven, P., Peterson, W. K., ... & Weimer, D. R. (1991). Magnetospheric boundary dynamics: DE 1 and DE 2 observations near the magnetopause and cusp. *Journal of Geophysical Research: Space Physics*, 96(A3), 3505-3522.
- Newell, P. T., & Meng, C. I. (1988). The cusp and the cleft/boundary layer: Low-altitude identification and statistical local time variation. *Journal of Geophysical Research: Space Physics*, 93(A12), 14549-14556.
- Newell, P. T., Meng, C. I., Sibeck, D. G., & Lepping, R. (1989). Some low-altitude cusp dependencies on the interplanetary magnetic field. *Journal of Geophysical Research: Space Physics*, 94(A7), 8921-8927.
- Nichols, J. D., Cowley, S. W. H., & McComas, D. J. (2006, March). Magnetopause reconnection rate estimates for Jupiter's magnetosphere based on interplanetary measurements at ~5AU. In *Annales Geophysicae* (Vol. 24, No. 1, pp. 393-406). Göttingen, Germany: Copernicus Publications.
- Nichols, J. D., Badman, S. V., Bagenal, F., Bolton, S. J., Bonfond, B., Bunce, E. J., ... & Yoshikawa, I. (2017). Response of Jupiter's auroras to conditions in the interplanetary medium as measured by the Hubble Space Telescope and Juno. *Geophysical Research Letters*, 44(15), 7643-7652.
- Nykyri, K., Cargill, P. J., Lucek, E., Horbury, T., Lavraud, B., Balogh, A., ... & Reme, H. (2004, July). Cluster observations of magnetic field fluctuations in the high-altitude cusp. In *Annales Geophysicae* (Vol. 22, No. 7, pp. 2413-2429). Copernicus GmbH.
- Nykyri, K., Otto, A., Adamson, E., Kronberg, E., & Daly, P. (2012). On the origin of high-energy particles in the cusp diamagnetic cavity. *Journal of Atmospheric and Solar-Terrestrial Physics*, 87, 70-81.
- Øieroset, M., Sandholt, P. E., Denig, W. F., & Cowley, S. W. H. (1997). Northward interplanetary magnetic field cusp aurora and high-latitude magnetopause reconnection. *Journal of Geophysical Research: Space Physics*, 102(A6), 11349-11362.
- Raines, J. M., D. J. Gershman, J. A. Slavin, T. H. Zurbuchen, H. Korth, B. J. Anderson, and S. C. Solomon (2014), Structure and dynamics of Mercury's magnetospheric cusp: Messenger measurements of protons and planetary ions, *J. Geophys. Res. Space Physics*, 119, 6587-6602.
- Reiff, P. H., Hill, T. W., & Burch, J. L. (1977). Solar wind plasma injection at the dayside magnetospheric cusp. *Journal of Geophysical Research*, 82(4), 479-491.
- Pitout, F., Escoubet, C. P., Klecker, B., & Dandouras, I. (2009, May). Cluster survey of the mid-altitude cusp—Part 2: Large-scale morphology. In *Annales Geophysicae* (Vol. 27, No. 5, pp. 1875-1886). Göttingen, Germany: Copernicus Publications.

- Pitout, F., & Bogdanova, Y. V. (2021). The polar cusp seen by Cluster. *Journal of Geophysical Research: Space Physics*, 126(9), e2021JA029582.
- Poh, G., Slavin, J. A., Jia, X., DiBraccio, G. A., Raines, J. M., Imber, S. M., ... & Solomon, S. C. (2016). MESSENGER observations of cusp plasma filaments at Mercury. *Journal of Geophysical Research: Space Physics*, 121(9), 8260-8285.
- Shi, Q. Q., Pu, Z. Y., Soucek, J., Zong, Q. G., Fu, S. Y., Xie, L., ... & Reme, H. (2009). Spatial structures of magnetic depression in the Earth's high-altitude cusp: Cluster multipoint observations. *Journal of Geophysical Research: Space Physics*, 114(A10).
- Slavin, J. A., DiBraccio, G. A., Gershman, D. J., Imber, S. M., Poh, G. K., Raines, J. M., ... & Solomon, S. C. (2014). MESSENGER observations of Mercury's dayside magnetosphere under extreme solar wind conditions. *Journal of Geophysical Research: Space Physics*, 119(10), 8087-8116.
- Smith, M. F., & Lockwood, M. (1996). Earth's magnetospheric cusps. *Reviews of Geophysics*, 34(2), 233-260.
- Spreiter, J. R., Summers, A. L., & Alksne, A. Y. (1966a). Hydromagnetic flow around the magnetosphere. *Planetary and Space Science*, 14(3), 223-253.
- Spreiter, J. R., Alksne, A. Y., & Abraham-Shrauner, B. (1966b). Theoretical proton velocity distributions in the flow around the magnetosphere. *Planetary and Space Science*, 14(11), 1207-1220.
- Stone, R. G., Pedersen, B. M., Harvey, C. C., Canu, P., Cornilleau-Wehrlin, N., Desch, M. D., ... & Zarka, P. (1992). Ulysses radio and plasma wave observations in the Jupiter environment. *Science*, 257(5076), 1524-1531.
- Sulaiman, A. H., Mauk, B. H., Szalay, J. R., Allegrini, F., Clark, G., Gladstone, G. R., ... & Bolton, S. J. (2022). Jupiter's low-altitude auroral zones: Fields, particles, plasma waves, and density depletions. *Journal of Geophysical Research: Space Physics*, 127(8), e2022JA030334.
- Tsyganenko, N. A., & Russell, C. T. (1999). Magnetic signatures of the distant polar cusps: Observations by Polar and quantitative modeling. *Journal of Geophysical Research: Space Physics*, 104(A11), 24939-24955.
- Winslow, R. M., C. L. Johnson, B. J. Anderson, H. Korth, J. A. Slavin, M. E. Purucker, and S. C. Solomon (2012), Observations of Mercury's northern cusp region with MESSENGER's Magnetometer, *Geophys. Res. Lett.*, 39, L08112.
- Woch, J., & Lundin, R. (1992). Magnetosheath plasma precipitation in the polar cusp and its control by the interplanetary magnetic field. *Journal of Geophysical Research: Space Physics*, 97(A2), 1421-1430.
- Zhang, B., Brambles, O., Lotko, W., Dunlap-Shohl, W., Smith, R., Wiltberger, M., & Lyon, J. (2013). Predicting the location of polar cusp in the Lyon-Fedder-Mobarry global magnetosphere simulation. *Journal of Geophysical Research: Space Physics*, 118(10), 6327-6337.
- Zhang, B., Delamere, P. A., Yao, Z., Bonfond, B., Lin, D., Sorathia, K. A., ... & Lyon, J. G. (2021). How Jupiter's unusual magnetospheric topology structures its aurora. *Science Advances*, 7(15), eabd1204.
- Zhou, X. W., Russell, C. T., Le, G., Fuselier, S. A., & Scudder, J. D. (2001). Factors controlling the diamagnetic pressure in the polar cusp. *Geophysical research letters*, 28(5), 915-918.

Alternatively, if the purpose of the paper is to show that the Juno data can be used to validate the Zhang et al simulation then this should be the expressed goal of the paper and not a claim of finding Jupiter's cusp region. In either case this reviewer would insist that several data examples of the phenomena and/or set of ion energy dispersion examples establishing the uniqueness of this proposed cusp example must be presented and a very consistent pattern identified that illustrates the existence of a cusp. If on the other hand, the goal of the paper is to validate the Zhang et al simulations then a series of data set examples of Juno observations must be presented that span the simulation regions of note and that are used to build a compelling case that the Zhang et al simulations represent reality.

Response #3

Many thanks to the referee for the detailed comments and clear guidance on how we can improve our manuscript. This study primarily aims to report Jupiter's cusp region through the latest Juno datasets, which was indeed inspired by the unusual magnetic configuration shown in the simulation in Zhang et al. (2021). The magnetic configuration revealed by Zhang et al. (2021) well explains Jupiter's auroral pictures that differ from those of Earth and Saturn. And magnetic structure pictures shown in Zhang et al. (2021) and Chen et al. (2023) can help our understanding of the observations.

We agree that more cusp cases are necessary in illustrating the existence of Jovian cusp. Following the suggestions by the referee, we have added more examples of cusp events exhibiting typical plasma characteristics, such as ion energy dispersion/reversed dispersion, to the revised manuscript and supplementary information. This enhancement underscores the consistency in cusp identification methods and strengthens the reliability of our findings. For the specific discussion and presentation of the cusp examples, please see **Response #2 section (2)** for details.

Reference

- Chen, J., Zhang, B., Lin, D., Delamere, P. A., Yao, Z., Brambles, O., ... & Lyon, J. G. (2023). Prediction of axial asymmetry in Jovian magnetopause reconnection. *Geophysical Research Letters*, 50(9), e2022GL102577.
- Zhang, B., Delamere, P. A., Yao, Z., Bonfond, B., Lin, D., Sorathia, K. A., ... & Lyon, J. G. (2021). How Jupiter's unusual magnetospheric topology structures its aurora. *Science Advances*, 7(15), eabd1204.

Furthermore, no clear definition and comparison of a cusp at Earth, Jupiter, or Saturn are introduced in a manner that allows a compelling metric to be developed which would verify the present conjecture. Given the content of the present paper, the authors have simply observed a likely reconnection event in a different place in Jupiter's magnetosphere, but have failed to convince this reviewer that this is a cusp (or to validate the Zhang et al. simulation) based on the existence of one data set.

Response #4

We thank the referee for the perceptive comment on the definition of cusp among Earth and giant planets. Establishing rigorous criteria for Jupiter cusp identification, along with a comparative analysis of cusp identification criteria for Earth, Saturn, and Jupiter, is essential

for this research. In alignment with the referee's suggestions, we have provided a clear definition and comparison of the identification criteria for cusps on Earth, Saturn, and Jupiter, drawing from previous studies. We believe that, following the reviewer's suggestion, we have now established clear and convincing identification criteria for Jupiter's cusp in the revised manuscript. For the specific discussion of cusp identification criteria, please see **Response #2 section (1)** for details.

Response to Referee #2's comments

We would like to extend our sincere appreciation to the referee for their valuable suggestions and insightful comments. Below are our point-by-point responses to the referee's remarks. The original comments of the referee are quoted *in italics* for convenience. Corresponding revisions have been made according to the referee's suggestions. The line numbers in our response refer to the highlighted draft.

Reviewer #2 (Remarks to the Author):

Xu et al., present Juno particle observations from high latitude dusk observations at Jupiter's magnetosphere which they identify as being cusp plasma, i.e. plasma from the shocked solar wind which has been injected upon the reconnection of a closed magnetospheric field with the interplanetary magnetic field. The key identifying feature which the authors use to identify the cusp plasma are plasma energies similar to the magnetosheath. If this observation was indeed of the cusp, then this would provide a significant contribution to our knowledge of the Jovian magnetosphere.

Response #1

We deeply appreciate the referee's thoughtful and constructive feedback on our manuscript. And we are grateful for the referee's positive remarks regarding the potential significance of our work. To further enhance the persuasiveness of our observations regarding Jupiter's cusp, we have significantly improved the manuscript. Our amendments are as follows:

- (1). We have clarified the criteria for cusp comparison among Earth, Saturn, and Jupiter, ensuring the identification of cusp events is both more rigorous and reliable.
- (2). We have incorporated additional typical cusp cases, selecting events that are more representative (e.g., longer duration and more distinct ion dispersion). These cases have been integrated into the main text and the supporting information, improving the limitations in the original manuscript related to the short duration and unclear ion dispersion of the single event.
- (3). We have provided a more detailed explanation regarding the location of the cusp event on the dusk side, supported by high-precision simulations in Chen et al. (2023) and Zhang et al. (2021).

The subsequent sections will elaborate on these three enhancements in detail.

(1) Criteria for the Identification of Earth's, Saturn's and Jupiter's Cusps.

As shown in **Table R1**, based on previous studies, the identification criteria for the cusps of Earth, Saturn, and Jupiter (inferred) are categorized into three levels of importance: level 1, level 2, and level 3, in descending order. Each criterion is elucidated and discussed sequentially below.

Importance and Reliability: Level 1 > 2 > 3

Identification of Cusp	Level 1 (key)				Level 2		Level 3		
	Inside Magnetopause	High Latitude Location	Magnetosheath-like Plasma/Electron	Magnetic Depression	Reconnection-Related Ion Dispersion	-	Whistler-mode Hiss Wave	Electron Pitch Angle Features	Other
Earth	Inside Magnetopause	High Latitude Location	Magnetosheath-like Plasma/Electron	Magnetic Depression	Reconnection-Related Ion Dispersion	-	Whistler-mode Hiss Wave	Electron Pitch Angle Features	Other
Saturn	Inside Magnetopause	High Latitude Location	Magnetosheath-like Plasma/Electron	-	Reconnection-Related Ion Dispersion	Magnetic Depression	Whistler-mode Hiss Wave	Electron Pitch Angle Features	Other
Jupiter (inferred)	Inside Magnetopause	High Latitude Location	Magnetosheath-like Plasma/Electron	-	Reconnection-Related Ion Dispersion	Magnetic Depression	Whistler-mode Hiss Wave	Electron Pitch Angle Features	Other

Table R1. Summary comparison of cusp identification criteria for Earth, Saturn and inferences of identification criteria for Jupiter's cusp.

- Level 1 identification criteria for cusp (**most important**).

- (a) Magnetosheath plasma features at high latitude inside the magnetosphere.

The magnetospheric cusp of a planet can be defined as part of the magnetosphere in the vicinity of the polar region at high magnetic latitudes/ invariant latitude, where a significant quantity of magnetosheath plasma is detected inside the magnetopause position (e.g., Cargill et al., 2005; Spreiter et al., 1966a; Smith and Lockwood et al., 1996). Defined by its nature and position, a cusp is an region where magnetosheath plasma and momentum can enter the magnetosphere. Therefore, the spacecraft's high-latitude positioning and the detection of magnetosheath plasma constitute the primary criteria for cusp identification., high-latitude location and magnetosheath plasma/electron features are listed as 'level 1' identification criteria.

At Earth, magnetosheath particles entering the cusp typically include low energy ions within the 500 eV–5 keV range and soft electrons around 50 eV (e.g., Lin et al., 1986; Pitout & Bogdanova, 2021). Low-energy magnetosheath-like electron distributions (10s ~ 100s eV enhancement) have been used to identify cusp events (in conjunction with the spacecraft's inside-magnetopause and high-latitude position) since in-situ observations in 1971 (Heikkila & Winningham, 1971; Frank, 1971), and this identification criterion has been used in cusp case reasearch for decades (e.g., Lin et al., 1986; Lockwood et al., 1993; Cargill et al. 2005). The distinction between Earth's cusp and other boundary layers has utilized the criterion of average electron spectrum enhancement, with levels $E_e < 220$ eV indicating magnetosheath-like conditions typical for the cusp, and enhancements from $220 < E_e < 600$ eV identifying other boundaries (e.g., cleft/LLBL, Newell et al., 1988, 1989). Thus, magnetosheath-like electrons at high latitudes within the magnetosphere constitute a dependable criterion for identifying cusp regions, a methodology that has also been applied to the identification of Saturn's cusp (~100 eV electron enhancement, Jasinski et al., 2014, 2016; Arridge et al., 2016). In contrast, sheath-like ion features are more variable and less reliable for identification purposes than electrons, due to the velocity filtering effects linked with reconnection and the intermittent nature of cusp-related reconnection (e.g., Lavraud et al., 2004). While electrons are also accelerated during reconnection, the velocity change is negligible (e.g., Smith & Lockwood, 1996).

- (b) Magnetic depression.

It should be noted that magnetic depression features are also very important for identifying cusp at Earth (e.g., Tsyganenko & Russell 1999; Dunlop et al. 2005; Zhang et al. 2013). The magnetic depression observed in the cusp region results from elevated plasma pressure and density due to the inflow of magnetosheath hot plasma, which is a magnetic turbulence feature that accompanies the magnetosheath inflow. In the cusp region, plasma thermal and magnetic pressures maintain a state of relative equilibrium. An increase in plasma thermal pressure leads to a significant reduction in magnetic pressure within the outer cusp, ensuring the maintenance of pressure equilibrium with the adjacent magnetospheric regions (e.g., Zhou et al., 2001; Shi et al., 2009). Lavraud et al. (2004) statistically demonstrated the presence of a diamagnetic cavity within the cusp region.

However, magnetic depression features are not always present in Saturn cusp events. Magnetic depression features are inconsistently observed in Saturn's cusp events. While some cases exhibit marked magnetic depressions akin to Earth's, others display weak or no magnetic field disturbances (e.g., see Figure 4, overview of Jan 2007, Arridge et al., 2016). Jasinski et al. (2017) reported that only two out of five winter cusp events exhibited magnetic depressions. Similarly, analysis of Juno data indicates that Jupiter's cusp events, as displayed in section (2), largely lack magnetic depression features. The absence of this feature for some cusp cases of giant planets remains unresolved. A plausible explanation considers the correlation between the plasma thermal pressure in the cusp and the solar wind dynamic pressure surrounding the planet. The solar wind dynamic pressure at the orbits of the giant planets (Jupiter at 5 AU, Saturn at 9 AU) is significantly lower than at Earth, while their magnetic field strengths in the cusp regions are comparable (10-100 nT). Consequently, the plasma from the magnetosheaths of these giant planets enters the cusp with generally lower thermal pressure relative to magnetic pressure, not necessarily inducing magnetic depression. This relationship is attributed to the solar wind's thermal pressure being minor compared to its ram pressure, with the latter being a major contributor to magnetosheath pressure (Lavraud et al., 2004), which, in turn, influences the thermal pressure of plasma entering the cusp. According to Spreiter et al. (1966b), magnetosheath pressure, a linear function of the solar wind Mach number squared (M^2), can be approximated as proportional to V_{sw}^2 , and magnetosheath density correlates with solar wind density. Therefore, within reasonable Mach number ranges, magnetosheath pressure correlates with ρV_{sw}^2 . Lavraud et al. (2004) conducted a statistical analysis on the ratio of plasma thermal pressure to solar wind dynamic pressure inside and outside the cusp, finding the magnitudes to be very comparable (see Figure 6b in it). Furthermore, Zhou et al. (2001) and Jasinski et al. (2017) statistically examined the correlation between cusp magnetic depression and solar wind dynamic pressure for Earth and Saturn, respectively, with the latter utilizing the Michigan Solar Wind Model. Their findings indicated that higher solar wind dynamic pressures lead to more pronounced cusp magnetic depressions. And studies on Mercury (Winslow et al., 2012; Raines et al., 2014; Slavin et al., 2014; Poh et al., 2016) have shown magnetic depression to be stronger than at Earth. These investigations collectively highlight the significant influence of the solar wind environment on cusp magnetic depression phenomena.

In summary, based on the cusp's definition and extensive observational reports, the presence of magnetosheath-like electron features at high latitudes well within the magnetopause is determined as the paramount criterion for identifying Jupiter's cusp. Magnetic depression, considered a key identification criterion for Earth, is not deemed a necessary feature for the giant planets. Therefore, it is classified as a Level 2 criterion.

- Level 2 identification criteria for cusp (less important but helpful).

The cusp, being the region where magnetosheath plasma and momentum enter the magnetosphere, exhibits plasma characteristics that are significantly influenced by magnetic reconnection. At Earth, the cusp's ion characteristics are affected by the magnetic reconnection process occurring at low latitude (Reiff et al., 1977; Maynard et al. 1991) or in high-latitude lobe region (Øieroset et al. 1997; Bosqued et al., 1985). Numerous studies have shown that, within Earth's cusp, ions often display dispersion (e.g., Reiff et al., 1977; Maynard et al. 1991; Pitout et al. 2009) or "reversed" dispersion patterns (e.g., Woch & Lundin, 1992; Øieroset et al. 1997), which have been used as an important feature in identify cusp. These patterns arise from the velocity filtering effect linked to acceleration during magnetic reconnection. The $E \times B$ drift causes high-energy ions to reach a given altitude before their slower counterparts, which arrive at the same altitude but at a different location, such as latitude. This time-of-flight effect, in combination with transverse convection, results in the well-known ion dispersion (e.g., Reiff et al., 1997; Lockwood & Smith, 1994). Different dispersion characteristics correspond to different solar wind conditions and reconnection locations (e.g., Woch & Lundin, 1992; Pitout et al. 2009). Normal and reversed ion dispersion features have also been used to identify Saturn's cusp (Jasinski et al., 2014; Jasinski et al., 2016; Arridge et al., 2016).

However, at Earth, due to the cusp's dynamic nature and the sporadic occurrence of magnetic reconnection, plasma flows within the cusp often exhibit intermittent features (e.g., Haerendel et al., 1978; Bosqued et al., 2005; Escoubet et al., 2006; Pitout & Bogdanova, 2021), which are affected by the solar wind and the spacecraft's position. Consequently, not all observations within the cusp show clear ion dispersion. Reports on Saturn's cusp indicate similar findings (Jasinski et al., 2016; Arridge et al., 2016). Thus, ion dispersion is classified as a Level 2 criterion, underlining its significance yet acknowledging it is not necessity for cusp identification. Furthermore, as outlined in section (b) regarding Level 1 criteria, magnetic depressions are also categorized under Level 2.

- Level 3 identification criteria for cusp (not necessary but can help).

At Earth, enhanced plasma waves ranging from 1 to 100s Hz—where the upper limit is the electron cyclotron frequency—are often utilized in conjunction with magnetic fields and plasma flows to identify the cusp (e.g., Chen et al., 1997, 1998; Fung et al., 1997). These auroral-like hiss waves are believed to consist of whistler mode emissions (Gurnett & Frank, 1978; Chen et al., 1998) and are thought to be produced by a Cerenkov radiation mechanism (Maggs, 1976). Similarly, auroral hiss has been detected in the very high-latitude magnetosphere of Jupiter (Gurnett et al., 1979), akin to the hiss observed over

Earth's auroral zones . It has been proposed that these emissions were generated by field-aligned electron beams (Gurnett et al., 1979), which may arise from reconnection-related processes. Stone et al. (1992) considered observations of enhanced auroral hiss near 100 Hz as potential indicators of Jupiter's cusp. Although less extensively studied at Saturn, it has also shown associations between auroral hiss and cusp regions. Auroral hiss Enhancements around 100 Hz have been observed within parts of the cusp, and in adjacent areas (Jasinski et al., 2014; Arridge et al., 2016).

In summary, auroral hiss can be used to assist in the identification of cusp, which can be generated by field-aligned electron beams produced by cusp-associated reconnection processes. However due to the sporadic nature of reconnection and the complexity of the cusp environment, auroral hiss is not consistently present in cusp regions, and thus it is classified as a Level 3 criterion, which can help identify cusp but is not necessary.

Similarly, field-aligned electron pitch angle features (enhancements near 0 and 180°), linked to reconnection processes, can facilitate the identification of the cusp on both Earth and Saturn (Smith & Lockwood, 1996; Jasinski et al., 2014; Arridge et al., 2016). However, there may be other properties of the electron pitch angle in the cusp, for example, enhancements around 90° due to local acceleration related to gradients in reconnected quasi-potential (Nykyri et al., 2012) and wave-particle interactions (Nykyri et al., 2004; Grison et al., 2005). Consequently, given the variable nature of electron pitch angle data within the cusp, this feature is also classified as a Level 3 criterion—helpful for identification but not very important.

(2) More Cusp Case Presentations.

This section displays all cusp cases (Figure R1-R6) featured in this study, all well meeting the Level 1 criteria for Jupiter. Moreover, the majority of these events also exhibit characteristics outlined in Levels 2 and 3 as listed in Table R1 (e.g., ion dispersion and hiss wave), with the notable exception of magnetic depression. Another possible explanation for the lack of magnetic depressions is the local time all cases occur close to the dusk side, away from local noon. Statistical studies of the Earth's cusp have demonstrated that the closer the local time is to noon, the more pronounced the magnetic depression (Zhou et al., 2001). Similarly, cusp events at Saturn showing magnetic depression features were observed near noon (Arridge et al., 2016; Jasinski et al., 2017).

Identification of Cusp	Level 1 (key)			Level 2		Level 3		
	Inside Magnetopause	High Latitude Location	Magnetosheath-like Plasma/Electron	Reconnection-Related Ion Dispersion	Magnetic Depression	Whistler-mode Hiss Wave	Electron Pitch Angle Features	Other
Jupiter (inferred)								
Jovian cusp case 1	√	√	√	√	-	√	-	-
Jovian cusp case 2	√	√	√	√	-	√	√	-
Jovian cusp case 3	√	√	√	√	-	√	-	-
Jovian cusp case 4	√	√	√	√	-	√	-	-
Jovian cusp case 5	√	√	√	√	-	√	-	-
Jovian cusp case 6	√	√	√	√	-	√	√	-

Table R2. The list which outlines how the criteria are met for the extended cusp cases. Due to the poor coverage of electron pitch angle data across most event observations, features such as field-aligned electrons are not discernible in the majority of events.

- Cases all with typical ion energy dispersion

This section presents all cusp cases featured in this study, all well meeting the Level 1 and ion dispersion (normal or reversed) criteria for Jupiter. Furthermore, all events also exhibit enhanced auroral hiss characteristics outlined in Level 3 as listed in **Table R1**. The identification of field-aligned electron features is challenging due to the absence of significant electron data near the pitch angles of 0 and 180 degrees in most events. The compliance of all events with the identification criteria is detailed in **Table R2**.

Two cases shown in **Figure R1** and **R2**, exemplifying typical ion energy dispersion and reversed ion energy dispersion, are elaborately discussed in the main text. The case in the original manuscript was replaced by **Figure R1** and **R2** due to its short duration (only half an hour) and less clear dispersion features. For the analysis of each event, refer to the revised manuscript and supplementary materials. Each case shown contains three parts, the case data, Juno's footprint distribution, and Juno's location.

Figure R1. Cusp example 1 showing clear typical ion dispersion (panel d). (a) R-Theta-Phi magnetic field components in JSS (Jupiter-De-Spun-Sun) coordinate; (b) The total magnetic field strength; (c) The electron energy spectrogram; Ion Energy spectrogram for protons (d) and heavy ions (e), where heavy ions represent ions with m/q in the range of 5 and 6428; (f) Pitch angle distribution for electrons which is normalized at each time unit within energy ranges of 0.3 to 32 keV; (g) Plasma wave observations in the frequency range 50 to 300 Hz. The different regions that the spacecraft passes through are marked with different colors at the top and separated by dashed lines. 'M' is the magnetosphere, 'C' is the cusp, 'BL' is the boundary layer. The red arrows and white dashed line in panel (d) show the dispersion. The yellow arrows in panel (g) indicate the enhanced auroral hiss features. The blue dashed line demarcates the cusp into two regions, labeled as 'a' and 'b', each characterized by different plasma properties. (h) Traced distribution of spacecraft footprints before and after cusp observation in Left-Handed system III coordinates. The red regions are the main ovals, and the colored lines are the Juno footprint trajectories before and after the cusp region observation (June 26th to April 28th, 2023). (i) The position of the spacecraft around cusp observations, red lines representing the time range of the cusp case.

Figure R2. Cusp example 2 showing clear reversed ion dispersion (panel d), using the same format as Figure R1.

Figure R3. Cusp example 3 showing clear normal ion dispersion (panel d), using the same format as Figure R1.

Figure R4. Example 4 of Cusp observations, using the same format as Figure R1.

Figure R5. Example 5 of cusp observations, using the same format as Figure R1.

Figure R6. Example 6 of cusp observations, using the same format as Figure R1, which is the event in the original manuscript.

(3) The Understanding of the Unexpected Duskside Location of Cusp.

The auroral morphologies observed on Jupiter, which differ from those on Earth and Saturn, suggest an unusual magnetic field configuration and cusp distribution that sets Jupiter apart from the traditional auroral pictures associated with Earth. Terrestrial auroral emissions are generated by the precipitation of energetic particles along magnetic field lines, as shown in Figure R7a (Eather & Mende, 1971; Eather, 1967). These emissions form an auroral oval that encircles the magnetic pole (Figure 1a). In contrast, although Saturn's magnetic fields are also predominantly dipolar, its auroral emissions display significant variations with complex morphologies (Gérard et al., 2006; Grodent et al., 2005; Carbary, 2012), exhibiting features characteristic of both Earth-like and Jupiter-like structures (Figures R7b-d). At Jupiter, the aurora present a distinctly different morphology and evolutionary pattern as depicted in Figure R7e (Clarke et al., 2012; Greathouse et al., 2021; Sulaiman et al., 2022), characterized by a highly dynamic yet generally bright polar cap aurora and contrasting dark “polar collar” regions.

Zhang et al. (2021) provide the only study to date that comprehensively explains these different auroral morphologies on Jupiter, whose insights into Jupiter's magnetic field structure have also established a crucial foundation for understanding the unusual cusp locations observed in this study. The formation and configuration of Jupiter's cusp can potentially be attributed to the solar wind environment around Jupiter and the asymmetrical configuration of Jupiter's rapidly rotating magnetosphere. Presented by the simulation result in Chen et al. (2023), the strong centrifugal force exerted during Jupiter's rapid rotation causes its magnetospheric configuration to appear "flatter," resulting in a greater presence of topological eastward/westward components, as shown in Figure R8. This flattened structure is suggested to be more sensitive to east-west solar wind-induced magnetic reconnection, as shown in Figure R9.

Furthermore, previous research indicates that both the interplanetary magnetic field (IMF) azimuthal angle (Ebert et al., 2014) and the clock angle (Nichols et al., 2006, 2017) around Jupiter are predominantly around $\pm 90^\circ$, suggesting that the solar wind near Jupiter is primarily east-west oriented. Shown by the simulation result in Chen et al. (2023), this orientation facilitates a unique coupling of the solar wind to Jupiter's magnetosphere, which exhibits a distinct magnetic field configuration, as depicted in Figure R9. This configuration allows for the existence of open magnetic field lines coupled to the solar wind on the dusk side inside the magnetosphere. It is within this portion of the magnetosphere that the cusp region can be observed, as shown in the boxes in Figures R9a and R9b.

Corresponding revisions and additions have been made in the revised manuscript, please see line 95 to 186, line 196 to 201, line 209 to 216, line 240 to 245 in the revised manuscript and Text S2, S3 and S5 in the Supplementary Information.

Figure R8. Comparison of (a) Earth's and (b) Jupiter's magnetic configuration using Chen et al. (2023) simulation results. The former exhibits dawn-dusk symmetry, whereas the latter demonstrates dawn-dusk asymmetry attributed to the effects of rapid rotation.

Figure R9. Jupiter's magnetic configuration under different typical (a) eastward / (b) westward solar wind conditions. (c) and (d) depict the zoomed-in magnetospheric structure at the location of the cusp events under different solar wind conditions using Chen et al. (2023) simulation results. The yellow spheres, labeled "event 1" and "event 2," denote the positions of the spacecraft during the pre-dusk cusp case 1 (**Figure R1**) and the post-dusk cusp case 2 (**Figure R2**), which are discussed in detail in the revised manuscript.

Reference

- Arridge, C. S., Jasinski, J. M., Achilleos, N., Bogdanova, Y. V., Bunce, E. J., Cowley, S. W., ... & Krupp, N. (2016). Cassini observations of Saturn's southern polar cusp. *Journal of Geophysical Research: Space Physics*, 121(4), 3006-3030.
- Bosqued, J. M., Sauvaud, J. A., Reme, H., Crasnier, J., Galperin, Y. I., Kovrazhkin, R. A., & Gladyshev, V. A. (1985). Evidence for ion energy dispersion in the polar cusp related to a northward-directed IMF. *Advances in space research*, 5(4), 149-153.
- Bosqued, J. M., Escoubet, C. P., Frey, H. U., Dunlop, M., Berchem, J., Marchaudon, A., ... & Balogh, A. (2005). Multipoint observations of transient reconnection signatures in the cusp precipitation: A Cluster-IMAGE detailed case study. *Journal of Geophysical Research: Space Physics*, 110(A3).
- Carbary, J. F. (2012). The morphology of Saturn's ultraviolet aurora. *Journal of Geophysical Research: Space Physics*, 117(A6).
- Cargill, P. J., Lavraud, B., Owen, C. J., Grison, B., Dunlop, M. W., Cornilleau-Wehrin, N., ... & Nykyri, K. (2005). Cluster at the magnetospheric cusps. *Space science reviews*, 118, 321-366.
- Chen, S. H., Boardsen, S. A., Fung, S. F., Green, J. L., Kessel, R. L., Tan, L. C., ... & Craven, J. D. (1997). Exterior and interior polar cusps: Observations from Hawkeye. *Journal of Geophysical Research: Space Physics*, 102(A6), 11335-11347.
- Chen, J., Fritz, T. A., Sheldon, R. B., Spence, H. E., Spjeldvik, W. N., Fennell, J. F., ... & Gurnett, D. A. (1998). Cusp energetic particle events: Implications for a major acceleration region of the magnetosphere. *Journal of Geophysical Research: Space Physics*, 103(A1), 69-78.
- Chen, J., Zhang, B., Lin, D., Delamere, P. A., Yao, Z., Brambles, O., ... & Lyon, J. G. (2023). Prediction of axial asymmetry in Jovian magnetopause reconnection. *Geophysical Research Letters*, 50(9), e2022GL102577.
- Clarke, J. T. (2012). Auroral processes on Jupiter and Saturn. *Auroral phenomenology and magnetospheric processes: Earth and other planets*, 197, 113-122.
- Dunlop, M. W., Lavraud, B., Cargill, P., Taylor, M. G. G. T., Balogh, A., Réme, H., ... & Marchaudon, A. (2005). Cluster observations of the cusp: magnetic structure and dynamics. *Surveys in Geophysics*, 26, 5-55.
- Eather, R. H. (1967). Auroral proton precipitation and hydrogen emissions. *Reviews of Geophysics*, 5(3), 207-285.
- Eather, R. H., & Mende, S. B. (1971). Airborne observations of auroral precipitation patterns. *Journal of Geophysical Research*, 76(7), 1746-1755.
- Ebert, R. W., Bagenal, F., McComas, D. J., & Fowler, C. M. (2014). A survey of solar wind conditions at 5 AU: A tool for interpreting solar wind-magnetosphere interactions at

- Jupiter. *Frontiers in Astronomy and Space Sciences*, 1, 4.
- Escoubet, C. P., Bosqued, J. M., Berchem, J., Trattner, K. J., Taylor, M. G. G. T., Pitout, F., ... & Fazakerley, A. (2006). Temporal evolution of a staircase ion signature observed by Cluster in the mid-altitude polar cusp. *Geophysical research letters*, 33(7).
- Frank, L. A. (1971). Plasma in the earth's polar magnetosphere. *Journal of Geophysical Research*, 76(22), 5202-5219.
- Fung, S. F., Eastman, T. E., Boardsen, S. A., & Chen, S. H. (1997). High-altitude cusp positions sampled by the Hawkeye satellite. *Physics and Chemistry of the Earth*, 22(7-8), 653-662.
- Gérard, J. C., Grodent, D., Cowley, S. W. H., Mitchell, D. G., Kurth, W. S., Clarke, J. T., ... & Coates, A. J. (2006). Saturn's auroral morphology and activity during quiet magnetospheric conditions. *Journal of Geophysical Research: Space Physics*, 111(A12).
- Greathouse, T., Gladstone, R., Versteeg, M., Hue, V., Kammer, J., Giles, R., ... & Vogt, M. F. (2021). Local time dependence of Jupiter's polar auroral emissions observed by Juno UVS. *Journal of Geophysical Research: Planets*, 126(12), e2021JE006954.
- Grison, B., Sahraoui, F., Lavraud, B., Chust, T., Cornilleau-Wehrin, N., Reme, H., ... & André, M. (2005, December). Wave particle interactions in the high-altitude polar cusp: a Cluster case study. In *Annales Geophysicae* (Vol. 23, No. 12, pp. 3699-3713). Göttingen, Germany: Copernicus Publications.
- Grodent, D., Gérard, J. C., Cowley, S. W. H., Bunce, E. J., & Clarke, J. T. (2005). Variable morphology of Saturn's southern ultraviolet aurora. *Journal of Geophysical Research: Space Physics*, 110(A7).
- Gurnett, D. A., & Frank, L. A. (1978). Plasma waves in the polar cusp: Observations from Hawkeye 1. *Journal of Geophysical Research: Space Physics*, 83(A4), 1447-1462.
- Gurnett, D. A., Kurth, W. S., & Scarf, F. L. (1979). Auroral hiss observed near the Io plasma torus. *Nature*, 280(5725), 767-770.
- Haerendel, G., Paschmann, G., Sckopke, N., Rosenbauer, H., & Hedgecock, P. C. (1978). The frontside boundary layer of the magnetosphere and the problem of reconnection. *Journal of Geophysical Research: Space Physics*, 83(A7), 3195-3216.
- Heikkila, W. J., & Winningham, J. D. (1971). Penetration of magnetosheath plasma to low altitudes through the dayside magnetospheric cusps. *Journal of Geophysical Research*, 76(4), 883-891.
- Jasinski, J. M., Arridge, C. S., Lamy, L., Leisner, J. S., Thomsen, M. F., Mitchell, D. G., ... & Waite, J. H. (2014). Cusp observation at Saturn's high-latitude magnetosphere by the Cassini spacecraft. *Geophysical Research Letters*, 41(5), 1382-1388.
- Jasinski, J. M., Arridge, C. S., Coates, A. J., Jones, G. H., Sergis, N., Thomsen, M. F., ... & Waite Jr, J. H. (2016). Cassini plasma observations of Saturn's magnetospheric cusp. *Journal of Geophysical Research: Space Physics*, 121(12), 12-047.
- Lavraud, B., Fedorov, A., Budnik, E., Grigoriev, A., Cargill, P. J., Dunlop, M. W., ... & Balogh, A. (2004, September). Cluster survey of the high-altitude cusp properties: a three-year statistical study. In *Annales Geophysicae* (Vol. 22, No. 8, pp. 3009-3019). Göttingen, Germany: Copernicus Publications.
- Lin, C. S., Burch, J. L., & Winningham, J. D. (1986). Near-conjugate observations of polar cusp electron precipitation using DE 1 and DE 2. *Journal of Geophysical Research: Space Physics*, 91(A10), 11186-11202.

- Lockwood, M., Denig, W. F., Farmer, A. D., Davda, V. N., Cowley, S. W. H., & Lühr, H. (1993). Ionospheric signatures of pulsed reconnection at the Earth's magnetopause. *Nature*, 361(6411), 424-428.
- Lockwood, M., & Smith, M. F. (1994). Low and middle altitude cusp particle signatures for general magnetopause reconnection rate variations: 1. Theory. *Journal of Geophysical Research: Space Physics*, 99(A5), 8531-8553.
- Maggs, J. E. (1976). Coherent generation of VLF hiss. *Journal of Geophysical Research*, 81(10), 1707-1724.
- Maynard, N. C., Aggson, T. L., Basinska, E. M., Burke, W. J., Craven, P., Peterson, W. K., ... & Weimer, D. R. (1991). Magnetospheric boundary dynamics: DE 1 and DE 2 observations near the magnetopause and cusp. *Journal of Geophysical Research: Space Physics*, 96(A3), 3505-3522.
- Newell, P. T., & Meng, C. I. (1988). The cusp and the cleft/boundary layer: Low-altitude identification and statistical local time variation. *Journal of Geophysical Research: Space Physics*, 93(A12), 14549-14556.
- Newell, P. T., Meng, C. I., Sibeck, D. G., & Lepping, R. (1989). Some low-altitude cusp dependencies on the interplanetary magnetic field. *Journal of Geophysical Research: Space Physics*, 94(A7), 8921-8927.
- Nichols, J. D., Cowley, S. W. H., & McComas, D. J. (2006, March). Magnetopause reconnection rate estimates for Jupiter's magnetosphere based on interplanetary measurements at ~5AU. In *Annales Geophysicae* (Vol. 24, No. 1, pp. 393-406). Göttingen, Germany: Copernicus Publications.
- Nichols, J. D., Badman, S. V., Bagenal, F., Bolton, S. J., Bonfond, B., Bunce, E. J., ... & Yoshikawa, I. (2017). Response of Jupiter's auroras to conditions in the interplanetary medium as measured by the Hubble Space Telescope and Juno. *Geophysical Research Letters*, 44(15), 7643-7652.
- Nykyri, K., Cargill, P. J., Lucek, E., Horbury, T., Lavraud, B., Balogh, A., ... & Reme, H. (2004, July). Cluster observations of magnetic field fluctuations in the high-altitude cusp. In *Annales Geophysicae* (Vol. 22, No. 7, pp. 2413-2429). Copernicus GmbH.
- Nykyri, K., Otto, A., Adamson, E., Kronberg, E., & Daly, P. (2012). On the origin of high-energy particles in the cusp diamagnetic cavity. *Journal of Atmospheric and Solar-Terrestrial Physics*, 87, 70-81.
- Øieroset, M., Sandholt, P. E., Denig, W. F., & Cowley, S. W. H. (1997). Northward interplanetary magnetic field cusp aurora and high-latitude magnetopause reconnection. *Journal of Geophysical Research: Space Physics*, 102(A6), 11349-11362.
- Raines, J. M., D. J. Gershman, J. A. Slavin, T. H. Zurbuchen, H. Korth, B. J. Anderson, and S. C. Solomon (2014), Structure and dynamics of Mercury's magnetospheric cusp: Messenger measurements of protons and planetary ions, *J. Geophys. Res. Space Physics*, 119, 6587-6602.
- Reiff, P. H., Hill, T. W., & Burch, J. L. (1977). Solar wind plasma injection at the dayside magnetospheric cusp. *Journal of Geophysical Research*, 82(4), 479-491.
- Pitout, F., Escoubet, C. P., Klecker, B., & Dandouras, I. (2009, May). Cluster survey of the mid-altitude cusp—Part 2: Large-scale morphology. In *Annales Geophysicae* (Vol. 27, No. 5, pp. 1875-1886). Göttingen, Germany: Copernicus Publications.

- Pitout, F., & Bogdanova, Y. V. (2021). The polar cusp seen by Cluster. *Journal of Geophysical Research: Space Physics*, 126(9), e2021JA029582.
- Poh, G., Slavin, J. A., Jia, X., DiBraccio, G. A., Raines, J. M., Imber, S. M., ... & Solomon, S. C. (2016). MESSENGER observations of cusp plasma filaments at Mercury. *Journal of Geophysical Research: Space Physics*, 121(9), 8260-8285.
- Shi, Q. Q., Pu, Z. Y., Soucek, J., Zong, Q. G., Fu, S. Y., Xie, L., ... & Reme, H. (2009). Spatial structures of magnetic depression in the Earth's high-altitude cusp: Cluster multipoint observations. *Journal of Geophysical Research: Space Physics*, 114(A10).
- Slavin, J. A., DiBraccio, G. A., Gershman, D. J., Imber, S. M., Poh, G. K., Raines, J. M., ... & Solomon, S. C. (2014). MESSENGER observations of Mercury's dayside magnetosphere under extreme solar wind conditions. *Journal of Geophysical Research: Space Physics*, 119(10), 8087-8116.
- Smith, M. F., & Lockwood, M. (1996). Earth's magnetospheric cusps. *Reviews of Geophysics*, 34(2), 233-260.
- Spreiter, J. R., Summers, A. L., & Alksne, A. Y. (1966a). Hydromagnetic flow around the magnetosphere. *Planetary and Space Science*, 14(3), 223-253.
- Spreiter, J. R., Alksne, A. Y., & Abraham-Shrauner, B. (1966b). Theoretical proton velocity distributions in the flow around the magnetosphere. *Planetary and Space Science*, 14(11), 1207-1220.
- Stone, R. G., Pedersen, B. M., Harvey, C. C., Canu, P., Cornilleau-Wehrlin, N., Desch, M. D., ... & Zarka, P. (1992). Ulysses radio and plasma wave observations in the Jupiter environment. *Science*, 257(5076), 1524-1531.
- Sulaiman, A. H., Mauk, B. H., Szalay, J. R., Allegrini, F., Clark, G., Gladstone, G. R., ... & Bolton, S. J. (2022). Jupiter's low-altitude auroral zones: Fields, particles, plasma waves, and density depletions. *Journal of Geophysical Research: Space Physics*, 127(8), e2022JA030334.
- Tsyganenko, N. A., & Russell, C. T. (1999). Magnetic signatures of the distant polar cusps: Observations by Polar and quantitative modeling. *Journal of Geophysical Research: Space Physics*, 104(A11), 24939-24955.
- Winslow, R. M., C. L. Johnson, B. J. Anderson, H. Korth, J. A. Slavin, M. E. Purucker, and S. C. Solomon (2012), Observations of Mercury's northern cusp region with MESSENGER's Magnetometer, *Geophys. Res. Lett.*, 39, L08112.
- Woch, J., & Lundin, R. (1992). Magnetosheath plasma precipitation in the polar cusp and its control by the interplanetary magnetic field. *Journal of Geophysical Research: Space Physics*, 97(A2), 1421-1430.
- Zhang, B., Brambles, O., Lotko, W., Dunlap-Shohl, W., Smith, R., Wiltberger, M., & Lyon, J. (2013). Predicting the location of polar cusp in the Lyon-Fedder-Mobarry global magnetosphere simulation. *Journal of Geophysical Research: Space Physics*, 118(10), 6327-6337.
- Zhang, B., Delamere, P. A., Yao, Z., Bonfond, B., Lin, D., Sorathia, K. A., ... & Lyon, J. G. (2021). How Jupiter's unusual magnetospheric topology structures its aurora. *Science Advances*, 7(15), eabd1204.
- Zhou, X. W., Russell, C. T., Le, G., Fuselier, S. A., & Scudder, J. D. (2001). Factors controlling the diamagnetic pressure in the polar cusp. *Geophysical research letters*, 28(5), 915-918.

The high energy cutoff labelled A and B are irrelevant to the analysis of the cusp dispersion (i.e., Figure 3 e and 4 d). The focus should be on the low-energy cutoff of the ions (i.e. C and whatever the lowest energy is below 'A'). The lowest energy ions observed at any particular time are expected to be from the reconnection site, since higher energy ions can enter from the magnetosheath along the open field line after reconnection has occurred and the field line has convected and be measured at the same time as the lowest energy. This will change your estimated energies which will affect your distance calculation.

Response #2

Many thanks to the referee for the insightful comments and for highlighting the significance of examining the low-energy cutoff of ions in our analysis of cusp dispersion. Following the referee's suggestions, we focused on low-energy cutoffs for each ion dispersion feature inside the cusp in the revised manuscript and reestimated the reconnection position distance. Using cusp case 4 proton spectrum in the supporting information as an example in Figure R10, we choose points A and B representing the lowest energy to calculate the reconnection distance. Accordingly, we have revised Figure 4 and the method part, please see line 288 to line 307 in the revised manuscript.

Figure R10. Proton energy spectrum of cusp case 4 in the supporting information and sketch, an example showing the selected low-energy cutoff points.

The authors use, what is assumed to be a magnetic field model by Zhang et al., (2021) to show schematics of a map of the magnetosphere and the location of Juno in the polar ionosphere. This is used as a way of attempting to persuade the reader that the spacecraft crossed through the open field region. Figure 2 shows that during the data timeseries, Juno does not significantly change location in Local time or Latitude. However, according to Figure 1c Juno travels towards the dayside and then goes to the nightside again, which is confusing.

Response #3

We thank the referee for pointing out the unclear clarification on the difference between the coordinate of the Juno footprint in Figure 1c and JSS (Jupiter-De-Spun-Sun) coordinate used for the observation case in the original manuscript. The schematics of a map of the magnetosphere and the location of Juno in the polar ionosphere are displayed in magnetic coordinate, whose center is the magnetic dipole axis (z-axis, as depicted in **Figure R11**). In comparison, JSS coordinate used in observation case is defined using Jupiter's spin axis (see **Figure R11**). Since there is a tilt angle $\sim 9.5^\circ$ between the spin axis and magnetic dipole axis in Jupiter, spacecraft's magnetic latitude (MLat) and magnetic local time will continue to change periodically (see the bottom panel of **Figure R11**) during rotations. Therefore, the discrepancy explains why, within the magnetic coordinate framework, Juno's footprint appears circular, even as its latitude and local time in JSS coordinates did not change significantly during the event. **Figure R12** illustrates Juno's position within the magnetic coordinate system directly on a polar plot, where its trajectory forms a circular shape. Consequently, this suggests that Juno's footprints would appear circular in the magnetic coordinate system, as depicted in Figure 1c of the original manuscript. We have clarified the distinctions between the coordinate systems used in the schematic and the observations in the revised manuscript, please see lines 201 to 206 and Text S1 in the Supplementary Information.

The Difference between the Latitudes and Local Times

Figure R10. The difference between the magnetic coordinate and the JSS coordinate, and the Latitude (Lat) - Magnetic Latitude (Mlat), Local Time (LT) – Magnetic Local Time (MLT) variation on April 15th, 2022.

Figure R11. Juno's location under magnetic coordinate directly shown in a polar plot on April 15th, 2022, which presents a 'circle-like' shape. Therefore, the footprint of Juno is suggested to be a 'circle-like' shape as shown in the Figure 1c sketch in the original manuscript which is different from that in the JSS coordinate.

REVIEWERS' COMMENTS

Reviewer #1 (Remarks to the Author):

The resubmitted paper is very thorough and addresses all of my previous concerns. The present version is a valuable contribution to the literature on the nature of the Jovian magnetosphere solar wind interaction. Thanks to the authors for all of the additional work they put in to improve the quality of the presented work.

Reviewer #2 (Remarks to the Author):

The authors have mostly adequately addressed my concerns.

2 leftover comments.

Figure 1h – the colors do not match the descriptions in the caption. Are the “red regions” really the “main ovals”? Is it not blue? Considering the timeseries shows Cusp 1, 2, 3, shown sequentially I do not understand how cusp 3 is before, in between and after 1 and 2.

Line 162, what radial distance at this location is the magnetopause expected? – you can't “confirm its location within the magnetosphere” if this information is not provided.

Response to Referee #1's comments

We would like to extend our sincere appreciation to the referee for their valuable comments. Below are our point-by-point responses to the referee's remarks. The original comments of the referee are quoted *in italics* for convenience. Corresponding revisions have been made according to the referee's suggestions. The line numbers in our response refer to the highlighted draft.

Reviewer #1 (Remarks to the Author):

The resubmitted paper is very thorough and addresses all of my previous concerns. The present version is a valuable contribution to the literature on the nature of the Jovian magnetosphere solar wind interaction. Thanks to the authors for all of the additional work they put in to improve the quality of the presented work.

Response

We deeply appreciate the referee for the positive feedback and for recognizing our efforts to improve the manuscript. Thanks very much for the referee's thorough review and support.

Response to Referee #2's comments

We would like to extend our sincere appreciation to the referee for their valuable comments. Below are our point-by-point responses to the referee's remarks. The original comments of the referee are quoted *in italics* for convenience. Corresponding revisions have been made according to the referee's suggestions. The line numbers in our response refer to the highlighted draft.

Reviewer #2 (Remarks to the Author):

*The authors have mostly adequately addressed my concerns.
2 leftover comments.*

Figure 1h – the colors do not match the descriptions in the caption. Are the “red regions” really the “main ovals”? Is it not blue?

Response

Many thanks to the referee for pointing out this typo. We have corrected “red regions” to “blue regions”.

Considering the timeseries shows Cusp 1, 2, 3, shown sequentially I do not understand how cusp 3 is before, in between and after 1 and 2.

Response

Thanks for the comment. The spacecraft footprints in Fig. 1h are displayed in Left-Handed system III coordinate which rotates with the planet and the spin period is about 10 hr. The interval between the observations of Cusp 3 and Cusp 1,2 is very long (~6 hrs). Moreover, the observation duration of Cusp 3 is also long (~6 hrs), resulting in the footprint during Cusp 3 appearing after one rotation period, overlapping with Cusp 1 and Cusp 2.

In the original figure, to present the footprints during Cusp 1 and Cusp 2 without being covered, we have plotted the footprints during Cusp 1 and Cusp 2 above those during Cusp 3. Consequently, the footprints during Cusp 3 appear before, between, and after those of Cusp 1 and Cusp 2.

We have improved Fig. 1h to ensure that the footprints during Cusp 3 look continuous and are not truncated by those during Cusp 1 and Cusp 2.

Line 162, what radial distance at this location is the magnetopause expected? – you can't “confirm its location within the magnetosphere” if this information is not provided.

Response

Many thanks for the comment. Following the referee's suggestion, we have added the expected magnetopause location information, improving the descriptions accordingly to “Given that Jupiter's magnetopause at 20LT was predicted to be roughly 130 R_J to 200 R_J (Joy et al., 2002), it can be confirmed that Juno's location was well within the magnetosphere.”

Reference

Joy, S. P., Kivelson, M. G., Walker, R. J., Khurana, K. K., Russell, C. T., & Ogino, T. (2002). Probabilistic models of the Jovian magnetopause and bow shock locations. *Journal of Geophysical Research: Space Physics*, 107(A10), SMP-17.